# Introducing new lightning schemes into the CHASER (MIROC) chemistry climate model

Yanfeng He[1], Hossain Mohammed Syedul Hoque[1], Kengo Sudo[1,2]

[1] Graduate School of Environment Studies, Nagoya University, Nagoya, 464-8601, Japan

[2] Japan Agency for Marine–Earth Science and Technology (JAMSTEC), 237-0061, Yokohama, Japan

*Correspondence to*: Yanfeng He (hyf412694462@gmail.com)

**Abstract.** The formation of nitrogen oxides ($NO_x$) associated with lightning activities (hereinafter designated as $LNO_x$) is a major source of $NO_x$. In fact, it is regarded as the dominant $NO_x$ source in the middle to upper troposphere. Therefore, improving the prediction accuracy of lightning and $LNO_x$ in chemical climate models is crucially important. This study implemented three new lightning schemes with the CHASER (MIROC) global chemical transport/climate model. The first lightning scheme is based on upward cloud ice flux (ICEFLUX scheme). The second one (the original ECMWF scheme), also adopted in the European Centre for Medium-Range Weather Forecasts (ECMWF) forecasting system, calculates lightning flash rates as a function of $Q_R$ (a quantity intended to represent the charging rate of collisions between graupel and other types of hydrometeors inside the charge separation region), convective available potential energy (CAPE), and convective cloud-base height. For the original ECMWF scheme, by tuning the equations and adjustment factors for land and ocean, a new lightning scheme named ECMWF-McCAUL scheme was also tested in CHASER. The ECMWF-McCAUL scheme calculates lightning flash rates as a function of CAPE and column precipitating ice. In the original version of CHASER (MIROC), lightning is initially parameterized with the widely used cloud top height scheme (CTH scheme). Model evaluations with lightning observations conducted using the Lightning Imaging Sensor (LIS) and Optical Transient Detector (OTD) indicate that both the ICEFLUX and ECMWF schemes simulate the spatial distribution of lightning more accurately on a global scale than the CTH scheme does. The ECMWF-McCAUL scheme showed the highest prediction accuracy for the global distribution of lightning. Evaluation by atmospheric tomography (ATom) aircraft observations (NO) and tropospheric monitoring instrument (TROPOMI) satellite observations ($NO_2$) shows that the newly implemented lightning schemes partially facilitated the reduction of model biases (NO and $NO_2$) typically within the regions where $LNO_x$ is the major source of $NO_x$ when compared using the CTH scheme. Although the newly implemented lightning schemes have a minor effect on the tropospheric mean oxidation capacity compared to the CTH scheme, they led to marked changes of oxidation capacity in different regions of the troposphere. Historical trend analyses of flash and surface temperatures predicted using CHASER (2001–2020) show that lightning schemes predicted an increasing trend of lightning or no significant trends, except for one case of the ICEFLUX scheme, which predicted a decreasing trend of lightning. The global lightning rates of increase during 2001–2020 predicted by the CTH scheme were 17.69%/°C and 2.50%/°C, respectively, with and without meteorological nudging. The un-nudged runs also included the short-term surface warming but without the application of meteorological nudging. Furthermore, the ECMWF schemes predicted a larger increasing trend of lightning flash rates under the short-term surface warming by a factor of 4 (ECMWF-McCAUL scheme) and 5 (original ECMWF scheme) compared to the CTH scheme without nudging. In conclusion, the three new lightning schemes improved global lightning prediction in the CHASER model. However, further research is needed to assess the reproducibility of trends of lightning over longer periods.

**Keywords**

lightning, lightning scheme, lightning $NO_x$, chemistry-climate model, lightning under climate change

## 1 Introduction

Nitric oxide (NO) can be formed during lightning activities. Also, NO can be oxidized quickly to nitrogen dioxide ($NO_2$). An equilibrium between NO and $NO_2$ can be reached during daytime. Those gases are known collectively as $NO_x$ (Finney et al., 2014). Actually, $LNO_x$ is estimated as contributing approximately 10% of the global $NO_x$ source. Regarded as the dominant $NO_x$ source in the middle to upper troposphere (Schumann and Huntrieser, 2007; Finney et al., 2016a), $NO_x$ is associated with many chemical reactions in the atmosphere. Most importantly, NO reacts with peroxy radical to reproduce OH radical. Photochemical dissociation of $NO_2$ engenders the production of ozone (Isaksen and Hov, 1987; Grewe, 2007). The primary oxidants in the atmosphere, which are OH radical and ozone, control the oxidation capacity of the atmosphere. Results of several studies have indicated that global-scale $LNO_x$ emissions are an important contributor to ozone and other trace gases, especially in the upper troposphere (Labrador et al., 2005; Wild, 2007; Liaskos et al., 2015). Consequently, $LNO_x$ influences atmospheric chemistry and global climate to a considerable degree (Schumann and Huntrieser, 2007; Murray, 2016; Finney et al., 2016b; Tost, 2017). However, large uncertainties remain in predicting lightning and $LNO_x$ in chemical climate models (Tost et al., 2007). Therefore, improving lightning prediction accuracy and quantifying $LNO_x$ in chemical climate models is crucially important for future atmospheric research.

Global chemical climate models (CCMs) such as CHASER (MIROC) (Sudo et al., 2002; Sudo and Akimoto, 2007; Watanabe et al., 2011) most often use the convective cloud-top height to parameterize the lightning flash rate (Price and Rind, 1992; Lamarque et al., 2013). The Earth System Models (ESMs) recently used in the sixth Coupled Model Intercomparison Project (CMIP6) all used the convective cloud-top height to calculate the lightning flash rates (Thornhill et al., 2021). Not only in global CCMs but the studies of $LNO_x$ in regional-scale models have also made significant progress in recent years (Heath et al., 2016; Kang et al., 2019a; Kang et al., 2019b; Kang et al., 2020).

The spaceborne Lightning Imaging Sensor (LIS) and Optical Transient Detector (OTD) lightning observation data (Cecil et al., 2014) are often utilized to evaluate the performance of different lightning schemes. A new lightning scheme proposed by Finney et al. (2014), which is based on upward cloud ice flux, has shown better spatial and temporal correlation coefficients and root mean square errors (RMSEs) than the cloud top height scheme compared against the LIS/OTD lightning observations. Another lightning scheme also showed more accurate lightning prediction than the cloud top height scheme, which is also adopted in the ECMWF forecasting system (Lopez, 2016). This lightning scheme uses $Q_R$ (a quantity intended to represent the charging rate of collisions between graupel and other types of hydrometeors inside the charge separation region), convective available potential energy (CAPE), and convective cloud-base height to compute the lightning flash rate (Lopez, 2016). The two new lightning schemes (Finney et al., 2014; Lopez 2016) mentioned above have only been evaluated in a few chemical transport and climate models. The new lightning schemes are expected to be evaluated and compared in more chemical transport and climate models, such as CHASER. To achieve better prediction accuracy for lightning and better quantification of $LNO_x$ in chemical climate models, comparing and optimizing the existing lightning schemes and evaluating them with various observation data are also important.

Lightning simulations are also fundamentally important in chemical climate model studies for predictions of atmospheric chemical fields and climate. Nevertheless, different lightning schemes respond very differently on decadal to multi-decadal time scales under global warming. Some lightning schemes such as those using cloud top height or CAPE × precipitation rate as a proxy for calculating lightning indicate that lightning increases concomitantly with increasing global average temperature. By contrast, other lightning schemes, such as those using convective mass flux or upward cloud ice flux as a proxy of lightning, indicate that lightning will decrease as the global average temperature increases (Clark et al., 2017; Finney et al., 2018). Several studies (Price and Rind 1994; Zeng et al., 2008; Jiang and Liao 2013; Banerjee et al., 2014;

Krause et al., 2014; Clark et al., 2017) have found 5–16% increases in lightning flashes per degree of increase in global
mean surface temperatures with the lightning scheme based on cloud top height. Over the contiguous United States
(CONUS), the CAPE × precipitation rate proxy predicted a 12 ± 5% increase in the CONUS lightning flash rate per degree
of global mean temperature increase (Romps et al., 2014). Compared to the findings reported by Romps et al. (2014), Finney
et al. (2020) found a relatively small response of lightning to climate change (2 % K$^{-1}$) over Africa using a cloud-ice-based
parametrisation for lightning. By contrast, Finney et al. (2018) found a 15% global mean lightning flash rate decrease with
the lightning scheme based on upward cloud ice flux in 2100 under a strong global warming scenario. Furthermore, a 2.0%
decrease in global mean lightning flashes per degree of increase in the global mean surface temperature with the lightning
scheme based on convective mass flux has been reported by Clark et al. (2017). Although it remains unclear which lightning
scheme is best suited to predicting future lightning (Romps, 2019), comparing different lightning schemes in different
chemical climate models is valuable for consideration of the sensitivity of lightning to global warming.

This study introduced three new lightning schemes into CHASER (MIROC). The first lightning scheme (Finney et al., 2014)
is based on upward cloud ice flux. The second one (Lopez, 2016), also adopted in the ECMWF forecasting system,
calculates lightning flash rates as a function of $Q_R$ (defined in Sect. 2.2), CAPE, and convective cloud-base height. In the
case of the second lightning scheme, by tuning the equations and adjustment factors based on a study reported by McCaul et
al. (2009), a new lightning scheme named ECMWF-McCAUL scheme was also tested for CHASER (MIROC). The
ECMWF-McCAUL scheme calculates lightning flash rates as a function of CAPE and column precipitating ice. Our study
conducted detailed evaluation of lightning and LNO$_x$ by LIS/OTD lightning observations, NASA/ATom aircraft
observations, and TROPOMI satellite observations. The effects of different lightning schemes on the atmospheric chemical
fields were evaluated. Also, 20-year (2001–2020) historical trend analyses of lightning densities and LNO$_x$ emissions
simulated by different lightning schemes were conducted. Based on the results, the effects of LNO$_x$ emissions during 2001–
2020 on tropospheric NO$_x$ and O$_3$ column trends were estimated and discussed.

Research methods, including the model description and experiment setup, are described in Sect. 2. In Sect. 3.1, the
evaluation of lightning schemes using LIS/OTD lightning observations is explained. In Sect. 3.2, LNO$_x$ emission simulation
by different lightning schemes is evaluated with aircraft and satellite measurements. Section 3.3 presents a discussion of the
effects of different lightning schemes on the atmospheric chemical fields. Historical trends of lightning simulated by
different lightning schemes are analyzed and discussed in Sect. 3.4. Section 3.5 discussed how LNO$_x$ emissions (2001–2020)
trends influence the tropospheric NO$_x$ and O$_3$ column trends. Section 4 presents the discussions and conclusions of this
study.
**2 Method**
**2.1 Chemistry-climate model**
The model used for this study is the CHASER (MIROC) global chemical transport and climate model (Sudo et al., 2002;
Sudo and Akimoto, 2007; Watanabe et al. 2011; Ha et al., 2021), which incorporates consideration of detailed chemical and
transport processes in the troposphere and stratosphere. CHASER calculates the distributions of 94 chemical species and
reflects the effects of 269 chemical reactions (58 photolytic, 190 kinetic, 21 heterogeneous). Its tropospheric chemistry
incorporates consideration of Non-Methane Hydrocarbons (NMHC) oxidation and the fundamental chemical cycle of O$_x$–
NO$_x$–HO$_x$–CH$_4$–CO. Its stratospheric chemistry simulates chlorine-containing and bromine-containing compounds,
chlorofluorocarbons (CFCs), hydrofluorocarbons (HFCs), carbonyl sulfide (OCS), and N$_2$O. Furthermore, it simulates the
formation of polar stratospheric clouds (PSCs) and heterogeneous reactions on their surfaces. CHASER is coupled to the

MIROC AGCM ver. 5.0 (Watanabe et al., 2011). Grid-scale large-scale condensation and cumulus convection (Arakawa–Schubert scheme) are used to simulate cloud/precipitation processes. Aerosol chemistry is coupled with the SPRINTARS aerosol model (Takemura et al., 2009), which is also based on MIROC.

For this study, horizontal resolution used is T42 (2.8° × 2.8°), with vertical resolution of 36 σ-p hybrid levels from the surface to approximately 50 km. The AGCM meteorological fields (u, v, T) simulated by MIROC were nudged towards the six-hourly NCEP FNL data (https://rda.ucar.edu/datasets/ds083.2/, last access: 6 December 2021). Anthropogenic precursor emissions such as $NO_x$, CO, $O_3$, $SO_2$, and VOCs were obtained from the HTAP-II inventory for 2008 (https://edgar.jrc.ec.europa.eu/dataset_htap_v2, last access: 6 December 2021), with biomass burning emissions from MACC-GFAS (Inness et al., 2013). The monthly soil $NO_x$ emissions used in CHASER (MIROC) are constant for each year and are derived from Yienger and Levy (1995).

## 2.2 Lightning NOₓ emission scheme

The lightning flash rate in CHASER is originally parameterized by cloud-top height (Price and Rind, 1992, 1993), with a "C-shaped" $NO_x$ vertical profile adopted (Pickering et al., 1998). The equations used to calculate the lightning flash rate by cloud-top height are

$$F_l = 3.44 \times 10^{-5} H^{4.9} \tag{1}$$

$$F_o = 6.2 \times 10^{-4} H^{1.73} \tag{2}$$

where $F$ represents the total flash frequency (fl. min$^{-1}$), $H$ stands for the cloud-top height (km), and subscripts $l$ and $o$ respectively denote the land and ocean (Price and Rind, 1992).

For this study, three new lightning schemes are implemented into CHASER (MIROC). One is based on upward cloud ice flux. It calculates the lightning flash rate by the following equations, as described by Finney et al. (2014).

$$f_l = 6.58 \times 10^{-7} \phi_{ice} \tag{3}$$

$$f_o = 9.08 \times 10^{-8} \phi_{ice} \tag{4}$$

Therein, $f_l$ and $f_o$ respectively represent the flash density (fl. m$^{-2}$ s$^{-1}$) over land and ocean. Also, $\phi_{ice}$ is the upward cloud ice flux at 440 hPa (kg$_{ice}$ m$_{cloud}^{-2}$ s$^{-1}$) as calculated using

$$\phi_{ice} = \frac{q \times \Phi_{mass}}{c}, \tag{5}$$

where $q$ denotes the specific cloud ice water content at 440 hPa (kg$_{ice}$ kg$_{air}^{-1}$), $\Phi_{mass}$ stands for the updraught mass flux at 440 hPa (kg$_{air}$ m$_{cell}^{-2}$ s$^{-1}$), and $c$ represents the fractional cloud cover at 440 hPa (m$_{cloud}^2$ m$_{cell}^{-2}$). The 440 hPa pressure level is chosen because it is a representative pressure level of fluxes in deep convective clouds (Finney et al., 2014). Moreover, Romps (2019) has proposed an alternative approach to applying the ICEFLUX scheme by using the upward cloud ice flux at 260-K isotherms instead of at 440 hPa isobars. As suggested by Romps (2019), the isotherm-alternative is more appropriate for climate change simulations because the charge separation zone will follow the isotherms instead of the isobars with climate change. The 260-K isotherm is chosen because it is close to the 440 hPa isobar based on a present-day tropical sounding and it lies within the mixed-phase regions of clouds (Romps, 2019). To distinguish the two different approaches to applying the ICEFLUX scheme, the isobar approach is abbreviated as ICEFLUX_P and the isotherm-alternative is abbreviated as ICEFLUX_T.

Another new lightning scheme, also adopted in the ECMWF forecasting system, calculates lightning flash rates as a function of the $Q_R$ (defined in equation 8), $CAPE$, and convective cloud-base height (Lopez, 2016) as

$$f_T = \alpha Q_R \sqrt{CAPE} \min(z_{base}, 1800)^2, \tag{6}$$

where $f_T$ is the total lightning flash density (fl. m$^{-2}$ s$^{-1}$), $z_{base}$ is the convective cloud-base height in m, $\alpha$ (fl. kg$^{-1}$ m$^{-3}$) is a
constant obtained after calibration against the LIS/OTD climatology, which is set to $1.11 \times 10^{-15}$ in this study. As explained by
Lopez (2016), the number 1800 used in equation (6) is a constraint to let the term $z_{base}$ remains constant after it exceeds
1800 m. Note that the equation (6) is standardized on base SI units. $CAPE$ (m$^2$ s$^{-2}$) is diagnosed from the following
equation.
$$CAPE = \int_{z_{LFC}}^{z_{w=0}} \max\left(g\frac{T_v^u - \overline{T_v}}{\overline{T_v}}, 0\right) dz \tag{7}$$
In that equation, $g$ is the constant of gravity. Also, $T_v^u$ and $\overline{T_v}$ respectively denote the virtual temperatures in the updraft and
the environment. The integral in equation (7) starts at the level of free convection $z_{LFC}$ and stops at the level at which
negative buoyancy is found ($w = 0$). Quantity $Q_R$ (kg m$^{-2}$) is intended to represent the charging rate of collisions between
graupel and other types of hydrometeors inside the charge separation region. It is empirically calculated as
$$Q_R = \int_{z_0}^{z_{-25}} q_{graup}(q_{cond} + q_{snow})\bar{\rho}dz, \tag{8}$$
where $z_0$ and $z_{-25}$ signify the heights (m) of the 0° and -25°C isotherms, and $q_{cond}$ denotes the mass mixing ratio of cumulus
cloud liquid water (kg kg$^{-1}$). The respective amounts of graupel ($q_{graup}$; kg kg$^{-1}$) and snow ($q_{snow}$; kg kg$^{-1}$) are computed
from the following equations for each vertical level of the model.
$$q_{graup} = \beta \frac{P_f}{\bar{\rho}V_{graup}} \tag{9}$$
$$q_{snow} = (1 - \beta)\frac{P_f}{\bar{\rho}V_{snow}} \tag{10}$$
In those equations, $P_f$ denotes the vertical profile of the frozen precipitation convective flux (kg m$^{-2}$ s$^{-1}$), $\bar{\rho}$ stands for the
environmental air density (kg m$^{-3}$), and $V_{graup}$ and $V_{snow}$ respectively express the typical fall speeds for graupel and snow set
to 3.0 and 0.5 m s$^{-1}$. The dimensionless coefficient $\beta$ is set as 0.7 for land and 0.45 for ocean to account for the observed
lower graupel contents over oceans.

For the original ECMWF scheme, by tuning the calculation equations based on findings reported by McCaul et al. (2009),
and the adjustment factors for land and ocean, the lightning prediction accuracy is improved further, as explained in Sect.
3.1. We named the new lightning scheme as ECMWF-McCAUL scheme, and it simulates the lightning flash rate by the
following equations.
$$f_l = \alpha_l Q_{Ra} CAPE^{1.3} \tag{11}$$
$$f_o = \alpha_o Q_{Ra} CAPE^{1.3} \tag{12}$$
Therein, $f_l$ and $f_o$ respectively denote the total flash density (fl. m$^{-2}$ s$^{-1}$) over land and ocean. Also, $\alpha_l$ and $\alpha_o$ are constants
(fl. s$^{1.6}$ kg$^{-1}$m$^{-2.6}$) obtained after calibration against LIS/OTD climatology, respectively, for land and ocean. For this
study, $\alpha_l$ and $\alpha_o$ are set respectively to $2.67 \times 10^{-16}$ and $1.68 \times 10^{-17}$. Then $CAPE$ is computed in the same way as the
original ECMWF scheme. In addition, $Q_{Ra}$ (kg m$^{-2}$) is a proxy for the charging rate resulting from the collisions between
graupel and hydrometeors of other types inside the charge separation region (from 0° to -25°C isotherm), as reported by
McCaul et al. (2009). Also, $Q_{Ra}$ represents the total volumetric amount of precipitating ice in the charge separation region,
calculated as
$$Q_{Ra} = \int_{z_0}^{z_{-25}} (q_{graup} + q_{snow} + q_{ice})\bar{\rho}dz, \tag{13}$$
where $q_{graup}$, $q_{snow}$, and $q_{ice}$ respectively represent the mass mixing ratios (kg kg$^{-1}$) of graupel, snow, and cloud ice. In
this study, $q_{graup}$ and $q_{snow}$ were computed respectively by equations (9) and (10). For the ECMWF-McCAUL scheme,
$V_{graup}$ and $V_{snow}$ are set respectively to 3.1 and 0.5 m s$^{-1}$. Then $q_{ice}$ was diagnosed using Arakawa–Schubert cumulus
parameterization.

**Table 1: Basic information of all lightning schemes assessed for this study**

| Abbreviation | Parameter | Remark |
|---|---|---|
| CTH (Price, C., & Rind, D., 1994) | Cloud top height | Originally used in CHASER (MIROC) |
| ICEFLUX (Finney et al., 2014) | Upward cloud ice flux at 440 hPa isobar (ICEFLUX_P) or at 260-K isotherm (ICEFLUX_T) | The 440 hPa level is used as a pressure level representative of fluxes in deep convective clouds |
| ECMWF-original (Lopez, 2016) | • $Q_R$ (Described in equation 8)<br>• CAPE<br>• Convective cloud-base height | Also adopted in the ECMWF forecasting system |
| ECMWF-McCAUL | • Column precipitating ice<br>• CAPE | Equations and adjustment factors are modified from the original ECMWF scheme. Equations are modified based on findings reported by McCaul (McCaul et al., 2009) |

Table 1 presents all the lightning schemes examined for this study. As described in this paper, the original ECMWF scheme and the ECMWF-McCAUL scheme are designated collectively as ECMWF schemes. Based on the recent studies, the intra-cloud (IC) lightning flashes are as efficient as the cloud-to-ground (CG) lightning flashes in $NO_x$ generation and the lightning $NO_x$ production efficiency ($LNO_x$ PE) is reported to be 100–400 mol per flash (Ridley et al., 2005; Cooray et al., 2009; Ott et al., 2010; Allen et al., 2019). Therefore, the $LNO_x$ PE values of IC and CG used in CHASER are set to the same value (250 mol per flash), which is the median of the commonly cited range of 100–400 mol per flash.

A fourth-order polynomial is used to calculate the proportion of total flashes that are cloud-to-ground ($p$) based on the cold cloud depth, as described in an earlier report (Price and Rind, 1993).

$$p = \frac{1}{64.09 - 36.54D + 7.493D^2 - 0.648D^3 + 0.021D^4}. \tag{14}$$

In that equation, $D$ represents the depth of cloud above the 0°C isotherms in kilometres.

**2.3 Observation data for model evaluation**

**2.3.1 Lightning observations**

The LIS/OTD gridded climatology datasets are used for this study, consisting of climatologies of total lightning flash rates observed using the Lightning Imaging Sensor (LIS) and spaceborne Optical Transient Detector (OTD): OTD aboard the MicroLab-1 satellite and LIS aboard the Tropical Rainfall Measuring Mission (TRMM) satellite (Cecil et al., 2014). Both sensors detect lightning by monitoring pulses of illumination produced by lightning in the 777.4 nm atomic oxygen multiplet above background levels. Both sensors, in low Earth orbit, view an Earth location for about 3 min as OTD passes overhead or for 1.5 min as LIS passes overhead. Actually, OTD and LIS circle the globe 14 times a day and 16 times a day, respectively. OTD collected data between +75 and -75° latitude from May 1995 through March 2000, whereas LIS observed between +38 and -38° latitude from January 1998 through April 2015. The product used throughout this paper is the LIS/OTD 2.5 Degree Low Resolution Time Series (LRTS). The LRTS includes the daily lightning flash rate on a 2.5° regular latitude–longitude grid from May 1995 through April 2015.

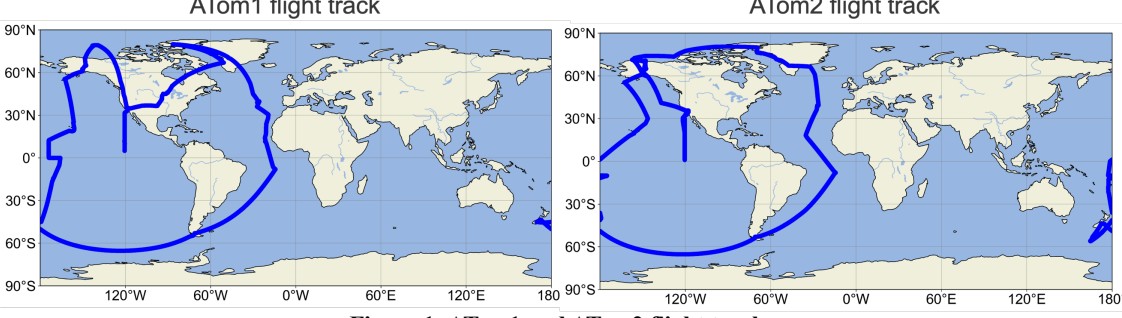

**Figure 1: ATom1 and ATom2 flight tracks.**

**2.3.2 Atmospheric tomography (ATom) aircraft observations**

To evaluate the $LNO_x$ emissions calculated by different lightning schemes, we used NO observation by the atmospheric tomography (ATom) aircraft missions (Wofsy et al., 2018). By deploying an extensive gas and aerosol payload on the

NASA DC-8 aircraft, ATom is designed to sample the atmosphere systematically on a global scale, performing continuous
profiling from 0.2 to 12 km altitude. Flights took place in each of the four seasons of 2016 through 2018. Since most of the
lightning occurs over land regions during summer, ATom1 (July–August 2016) and ATom2 (January–February 2017) were
used to evaluate LNO$_x$ emissions (corresponding to summer in the northern and southern hemispheres, respectively). Both
ATom1 and ATom2 originate from the Armstrong Flight Research Center in Palmdale, California, USA, fly north to the
western Arctic, south to the South Pacific, east to the Atlantic, north to Greenland, and return to California across central
North America. Figure 1 exhibits the respective flight tracks of ATom1 and ATom2. To evaluate the model simulated NO
against the ATom observations, we have sampled the specific flight track and timings from the modelled data.

### 241    2.3.3 TROPOMI satellite observations

Tropospheric Monitoring Instrument (TROPOMI) is the payload on-board the Sentinel-5 Precursor (S5P) satellite of the
European Space Agency (ESA), which was launched in October 2017. TROPOMI has been providing observations of
important atmospheric pollutants (NO$_2$, O$_3$, CO, CH$_4$, SO$_2$, CH$_2$O) with an unprecedented horizontal resolution of approx. 7
$\times$ 3.5 km$^2$ since August 2017 (changed to 5.5 $\times$ 3.5 km$^2$ after August 2019). The data used in this study is the TROPOMI
level-2 offline (OFFL) tropospheric NO$_2$ columns in 2019. The product version is 1.0.0 from 2019-01-01 to 2019-03-20 and
updated to 1.1.0 from 2019-03-21 to 2019-12-31. For the direct comparisons between TROPOMI level-2 products with
CHASER results, the following procedures were conducted to pre-process the TROPOMI data and CHASER modelled
fields.
1. The TROPOMI retrievals with quality assurance (QA) values of $\geqq 0.75$ were selected.
2. Horizontally, the TROPOMI data (tropospheric NO$_2$ columns, temperatures, pressures, averaging kernels) were
interpolated to the CHASER 2.8° $\times$ 2.8° grid.
3. The modelled results were sampled based on the TROPOMI overpass time. The CHASER tropospheric NO$_2$ columns
were calculated by using the sampled modelled results, the averaging kernels retrieved from the TROPOMI retrievals, and
the temperature and pressure profiles provided by TROPOMI retrievals. The averaging kernels are applied to each layer of
the CHASER outputs following the equation (16).
4. The pre-processed data described above were used to produce the monthly averaged data.

### 258    2.3.4 OMI satellite observations

Ozone Monitoring Instrument (OMI) is a key instrument onboard NASA's Aura satellite for measuring criteria pollutants
such as O$_3$, NO$_2$, SO$_2$, and aerosols. OMI has been providing observations with spatial resolution varying from 13 km $\times$ 25
km to 26 km $\times$128 km since October 2004 (Goldberg et al., 2019). The NO$_2$ product used in this study is the level-3 daily
global gridded (0.25° $\times$ 0.25°) Nitrogen Dioxide product (OMNO2d) (Nickolay et al. 2019). The O$_3$ product used in this
study is the monthly mean tropospheric column O$_3$ product developed from OMI in combination with Aura Microwave
Limb Sounder (MLS) with the detailed method described by Ziemke et al. (2006).

### 265    2.4 Experiment setup

For this study, all the introduced lightning schemes were implemented into CHASER (MIROC). Six sets of experiments
were conducted for this study and the detailed settings of all experiments are presented in Table 2. For each set of
experiments, the same initial conditions and chemical emissions were used except for LNO$_x$ emissions. The set of
experiments that applied meteorological nudging also has the same meteorological conditions. The monthly varying soil NO$_x$
emissions used are constant each year for all experiments derived from Yienger and Levy (1995). All experiments used the
"backward C-shaped" LNO$_x$ vertical profile (Ott et al., 2010). The LNO$_x$ PE values of IC and CG used in all experiments are
set to the same value (250 mol per flash), which is based on the recent literature (Ridley et al., 2005; Cooray et al., 2009; Ott
et al., 2010; Allen et al., 2019). It is noteworthy that there still exist large uncertainties in determining the $LNO_x$ PE values
(Allen et al., 2019; Bucsela et al., 2019) and the choice of different $LNO_x$ PE values may influence the simulated $LNO_x$
emissions and chemical fields. A more sophisticated parametrisation of $LNO_x$ PE values needs to be implemented and
verified in the chemistry-climate models in future research.

The first set of experiments was conducted for the years of 2001–2020. It was used to evaluate the distribution of the
lightning flash rate against LIS/OTD lightning observations and to derive the historical lightning trend. The second set of
experiments is the same as the first set of experiments, but uses daily mean $LNO_x$ emission rates of 2001 calculated using
lightning schemes for each year. This set of experiments is used to produce results for comparison with those of the first set
of experiments to estimate the effects of $LNO_x$ emission trends on tropospheric $NO_x$ and $O_3$ column trends. The third set of
experiments gives results for 2011–2020. These experiments are used to estimate the effects of different lightning schemes
on atmospheric chemical fields. To normalize the different annual $LNO_x$ emission amounts by different lightning schemes,
temporally and spatially uniform adjustment factors were applied to adjust the mean $LNO_x$ production (2011–2020) to 5.0
TgN yr$^{-1}$. Note the 10-years (2011–2020) mean $LNO_x$ production was adjusted to 5.0 TgN yr$^{-1}$ but the $LNO_x$ production in
each year is not exactly 5.0 TgN yr$^{-1}$. This adjustment was achieved by first conducting the simulations without any
adjustment and the 2011–2020 mean $LNO_x$ production ($P_{LNO_x}$) was calculated, then the corresponding adjustment factor
($adj\_factor$) can be calculated by using the following equation.
$$adj\_factor = \frac{5.0}{P_{LNO_x}} \qquad (15)$$
Similarly, we also adjusted the $LNO_x$ emissions in the fourth to the sixth sets of experiments to 5.0 TgN yr$^{-1}$. The fourth set
of experiments is for 2016, with the fifth set for 2017. These two sets of experiments were conducted to compare model
results with ATom1 and ATom2 aircraft observations. The sixth set of experiments is for 2019. It is conducted to evaluate
model results using TROPOMI satellite observations.

**Table 2: All experiments in this study**

| Number | 1st | | 2nd | | 3rd | 4th | 5th | 6th |
|---|---|---|---|---|---|---|---|---|
| Period | 2001–2020 | 2001–2020 | 2001–2020 | 2001–2020 | 2011–2020 | 2016 | 2017 | 2019 |
| Nudging | On | Off[a] | On | Off | On | On | On | On |
| $LNO_x$ emissions | Interactively calculated[b] | Interactively calculated | Fixed to 2001 | Fixed to 2001 | Interactively calculated | Interactively calculated | Interactively calculated | Interactively calculated |
| Adjusted to 5.0 TgN yr$^{-1}$ | No | No | No | No | Yes | Yes | Yes | Yes |
| Climate[c] | 2001-2020 (RCP4.5) | 2001-2020 (RCP4.5) | 2001-2020 (RCP4.5) | 2001-2020 (RCP4.5) | 2011-2020 (RCP4.5) | 2016 (RCP4.5) | 2017 (RCP4.5) | 2019 (RCP4.5) |
| Anthropogenic emissions | HTAP-II (2008) for all years | | | | | | | |
| Soil $NO_x$ emissions | Monthly varying values but constant for each year derived from Yienger and Levy (1995) | | | | | | | |
| Biomass burning emissions | MACC (2001-2020) | MACC (2001-2020) | MACC (2001-2020) | MACC (2001-2020) | MACC (2011-2020) | MACC (2016) | MACC (2017) | MACC (2019) |

[a]Nudging off means the meteorological fields (u, v, T) are free-running instead of nudging towards the NCEP FNL data.
[b]$LNO_x$ is interactively calculated by using different lightning schemes.
[c]The climate change is simulated by prescribed SST/sea ice fields and prescribed varying concentrations of GHGs ($CO_2$,
$N_2O$, methane, chlorofluorocarbons – CFCs – and hydrochlorofluorocarbons – HCFCs) utilized only in the radiation scheme.
The SST/sea ice fields are obtained from the HadISST dataset (Rayner et al., 2003).

## 3 Results and Discussion

### 3.1 Evaluation of the lightning schemes

As investigated by Finney et al. (2014), 5 years data are necessary and appropriate to produce a lightning climatology. Therefore, model results with nudging (2007-2011) were evaluated against the climatological lightning distributions of LIS (2007-2011) within ±38° latitude and LIS/OTD (1996-2000) within a broader range of ±75° latitude. We have evaluated the potential uncertainties associated with the inconsistency of the time period of simulated lightning and observed lightning (2007-2011 and 1996-2000). The statistical analysis between LIS (2007-2011) and LIS/OTD (1996-2000) within ±38° latitude exhibits an extremely high spatial correlation coefficient (R=0.99) and relatively small relative bias (0.65%), which supports the reasonability of comparing model results with the observation data within different time range.

The distribution of lightning observed by LIS/OTD and simulated by CHASER (MIROC) with different lightning schemes is depicted in Fig. 2. Figure 2 shows that lightning over the ocean is not well reproduced by the original CTH scheme. Actually, it is improved considerably by the new lightning schemes. Compared with the CTH scheme, the original ECMWF scheme better represents the lightning distribution in South Asia including the Indian region. The ECMWF schemes and the ICEFLUX_P scheme reduced negative biases in North America compared to the CTH scheme. In Australia, the ECMWF schemes better simulate the horizontal distribution of lightning. All lightning schemes failed to capture the worldwide maximum value found over the Congo Basin, although all lightning schemes captured the active region in central Africa.

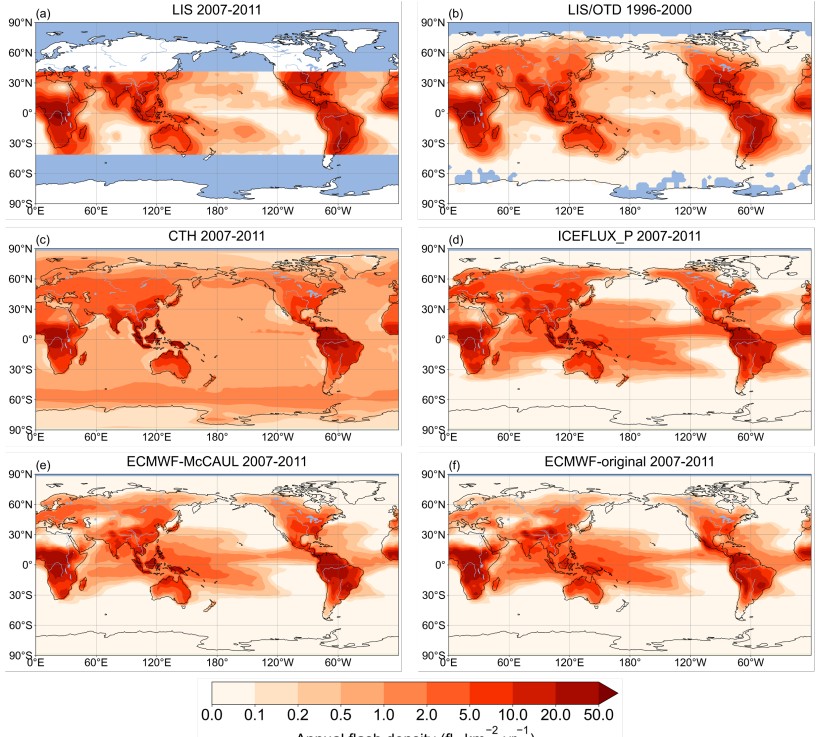

**Figure 2: Annual mean lightning flash densities from (a) LIS satellite observations spanning 2007–2011, (b) LIS/OTD satellite observations spanning 1996-2000 but with a wider range, (c) the CTH scheme in 2007–2011, (d) the ICEFLUX_P scheme in 2007–2011, (e) the ECMWF-McCAUL scheme in 2007–2011, and (f) the original ECMWF scheme in 2007–2011.**

To directly estimate the prediction accuracy of all lightning schemes, the Taylor diagrams are displayed in Fig. 3. In Fig. 3a, the overall prediction accuracy of the ICEFLUX_P and original ECMWF schemes evaluated against the LIS 2007-2011 lightning climatology is slightly improved compared to the CTH scheme. This improvement is more obvious when considering land and ocean separately (Figs. 3b-c). In the case of Figs. 3a-c, the ECMWF-McCAUL scheme has shown the best prediction accuracy among all lightning schemes. In Fig. 3d, comparison of the annual mean lightning flash rate of LIS/OTD 1996–2000 and the CHASER calculation for 2007–2011 yields spatial correlation coefficients of 0.80 and 0.79 for

the ICEFLUX_P and original ECMWF schemes, respectively, which are slightly higher than that found for the CTH scheme (0.78). The overall RMSE of the ICEFLUX_P scheme is 3.32 fl. km$^{-2}$ yr$^{-1}$, which is slightly less than that of the CTH scheme of 3.44 fl. km$^{-2}$ yr$^{-1}$. Among all lightning schemes, the ECMWF-McCAUL scheme exhibits the highest spatial correlation coefficient (0.83) and the lowest RMSE (3.20 fl. km$^{-2}$ yr$^{-1}$) as depicted in Fig. 3d. As displayed in Fig. 2, the prediction accuracy of lightning over the ocean is significantly improved, which can also be verified in Fig. 3f.

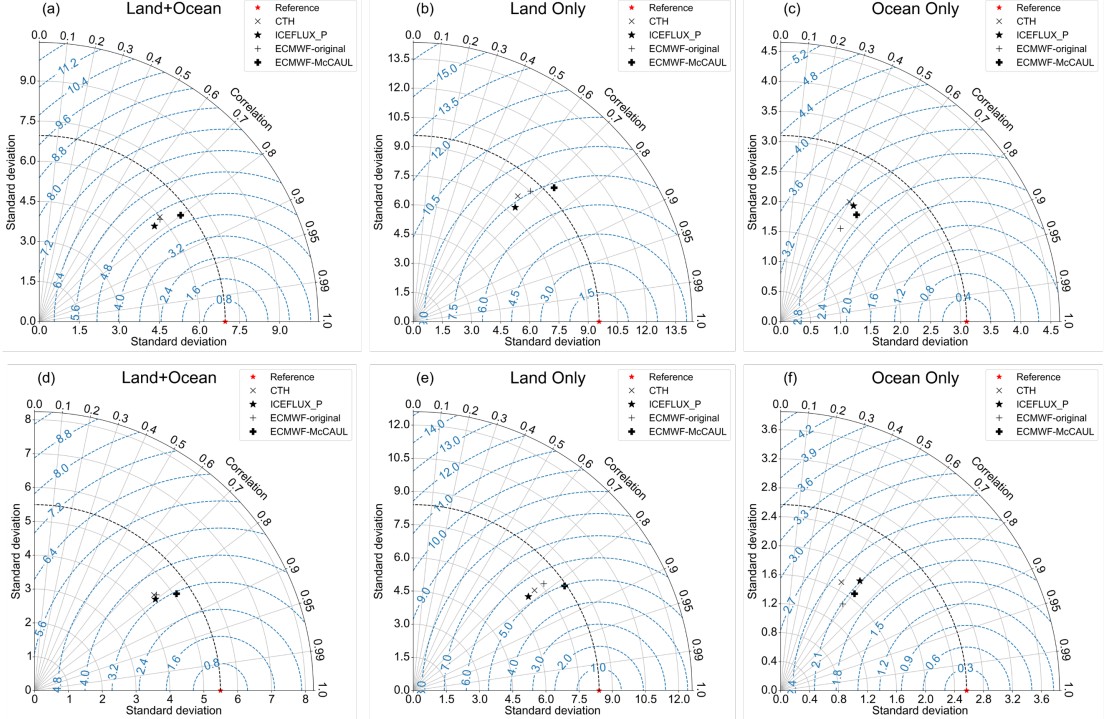

**Figure 3: Taylor diagram showing the prediction accuracy of various lightning schemes in 2007–2011 simulations compared to the LIS 2007-2011 lightning climatology (a-c) and the LIS/OTD 1996–2000 lightning climatology (d-f).**

To estimate whether the improvement of prediction accuracy discussed in Fig. 3 is significant, a significant test is conducted by considering the uncertainties in the LIS/OTD observations. Based on the uncertainties in the LIS/OTD observations, the probability density distributions (PDDs) of spatial correlation coefficients (R) and RMSE between the model and observations are derived by using a Monte Carlo method and displayed in Fig. 4. The uncertainties in the LIS/OTD observations are determined based on the uncertainties of the instrument bulk flash detection efficiency of LIS ($88 \pm 9\%$) and OTD ($54 \pm 8\%$) (Boccippio et al., 2002). The R and RMSE shown in Fig. 4 are all normally distributed which is determined by the Kolmogorov–Smirnov test. Based on the probability density functions of R and RMSE derived from Fig. 4, the order of R between the model and observations is estimated to be ECMWF-McCAUL > ICEFLUX_P > ECMWF-original > CTH with a confidence limit larger than 99.9%. Moreover, the order of RMSE between the model and observations is estimated to be ECMWF-McCAUL < ICEFLUX_P < ECMWF-original and CTH with a confidence limit larger than 95%. According to the significant test described above, we can conclude that the newly implemented lightning schemes have improved the lightning prediction accuracy compared to the original CTH scheme.

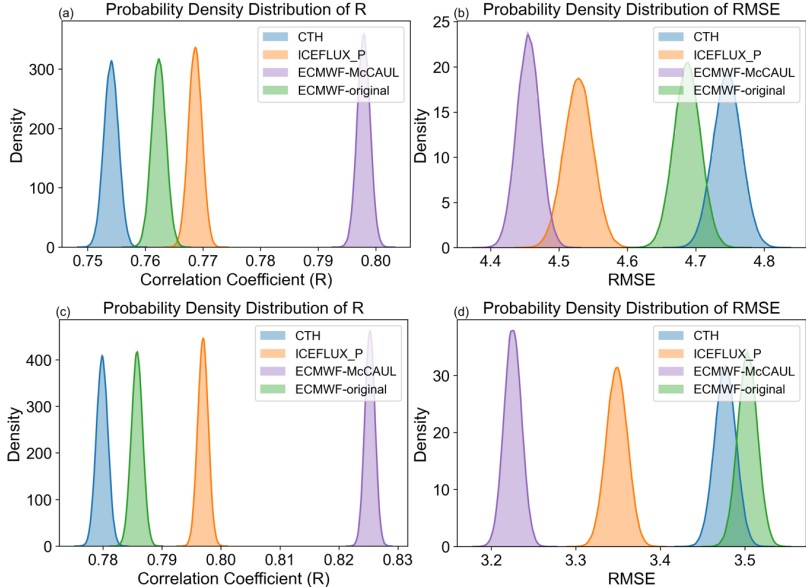

351

**Figure 4: The probability density distributions (PDDs) of spatial correlation coefficients (R) and RMSE between the model and LIS/OTD lightning observations. Figures 4(a–b) show the PDDs obtained between LIS lightning climatology (2007-2011) and the model outputs (2007-2011) within ±38° latitude. Figures 4(c–d) show the PDDs obtained between LIS/OTD lightning climatology (1996-2000) and the model results (2007-2011) within ±75° latitude.**

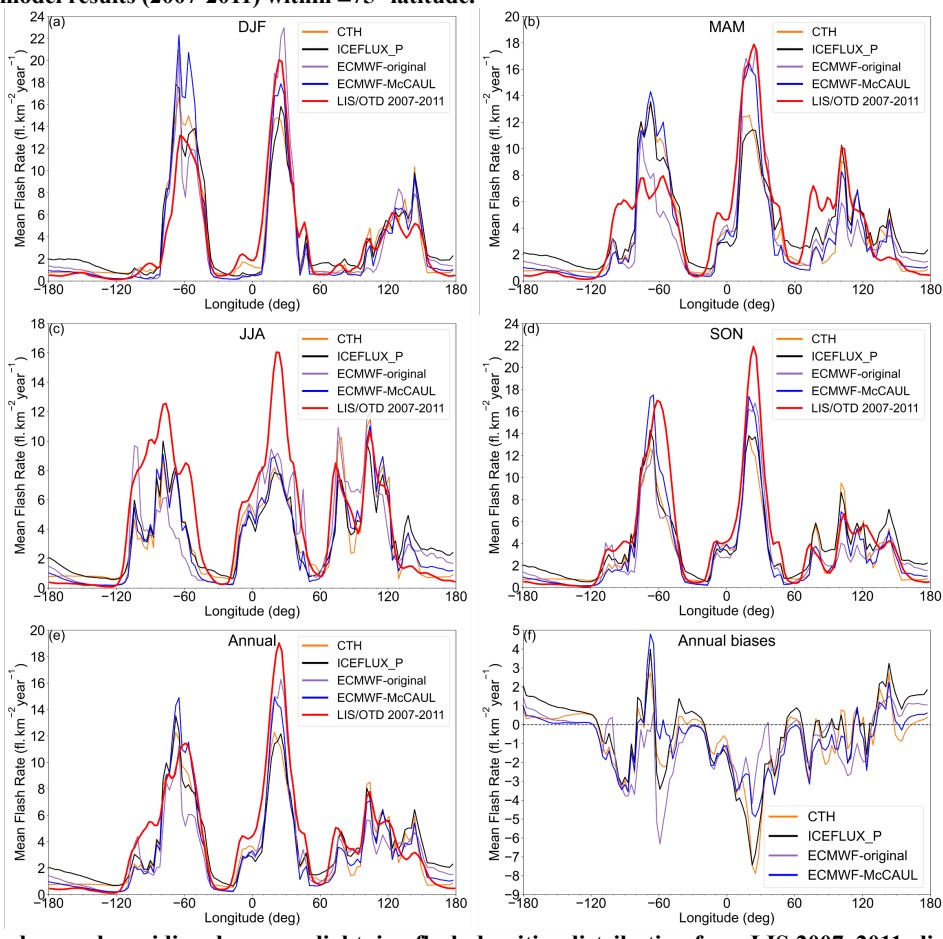

356

**Figure 5: Seasonal and annual meridional average lightning flash densities distribution from LIS 2007–2011 climatology (red line) and from simulation results (2007–2011) obtained using different lightning schemes. The meridional average is only taken within the LIS viewing region of ±38° latitude. The biases (model-obs.) in Fig. 5e are also portrayed in Fig. 5f.**

Figure 5 displays a comparison of seasonal and annual meridional average lightning flash densities from simulations (2007–2011) and LIS satellite observations (2007–2011). As Fig. 5 shows, the pairs of curves are usually in good agreement, even though the annual plot (Fig. 5e) highlights the underestimation which occurs for Africa (from 0 degrees to 30 degrees east) and North America (from 80 degrees west to 120 degrees west). The ECMWF schemes have made improvements within Africa. Also, the ICEFLUX_P and the original ECMWF schemes have slightly reduced the biases over North America. A

noticeable underestimation over the Americas in JJA and overestimation in MAM can be observed respectively in Figs. 5c
and 5b. Lightning densities over Africa are generally underestimated to varying degrees in different seasons, with the
greatest underestimation occurring in JJA (Fig. 5c). Lightning densities over Asia (from 60 degrees east to 120 degrees east)
are slightly underestimated in MAM (Fig. 5b). The ICEFLUX_P scheme has reduced the biases.

Figure 6 is the same as Fig. 5, but for the zonal mean distributions. The curves of the model results and the observation
results in Fig. 6 generally show good agreement. Figure 6f shows that, overall, the ICEFLUX_P and the ECMWF-McCAUL
schemes slightly overestimated the lightning densities near the equator (10°S–10°N). All lightning schemes underestimated
the lightning densities within 15°N–38°N and 20°S–38°S. Figure 6f also shows that the ICEFLUX_P scheme has reduced
the biases within 10°N–38°N and 15°S–38°S compared to the CTH scheme. In DJF (Fig. 6a), all lightning schemes
overestimated the flash densities over the low latitude regions but slightly underestimated the flash densities over the middle
latitude regions in the Southern Hemisphere. In MAM (Fig. 6b), lightning densities are overestimated near the equator and
underestimated over 15°N–38°N and 15°S–38°S by all lightning schemes to varying degrees. In JJA (Fig. 6c), noticeable
overestimation around 10°N by the original ECMWF scheme is apparent. Moreover, the CTH and the original ECMWF
schemes respectively facilitated reduction of model biases over 15°S–38°S and 15°N–38°N. As Fig. 6d shows, the model-
predicted lightning maximum value is shifted approximately 15 degrees to the north in SON compared to the lightning
observations. Figure 6d also shows that all lightning schemes underestimated the lightning densities over 15°N–38°N and
0°–38°S. The ICEFLUX_P scheme has shown improvement over these regions.

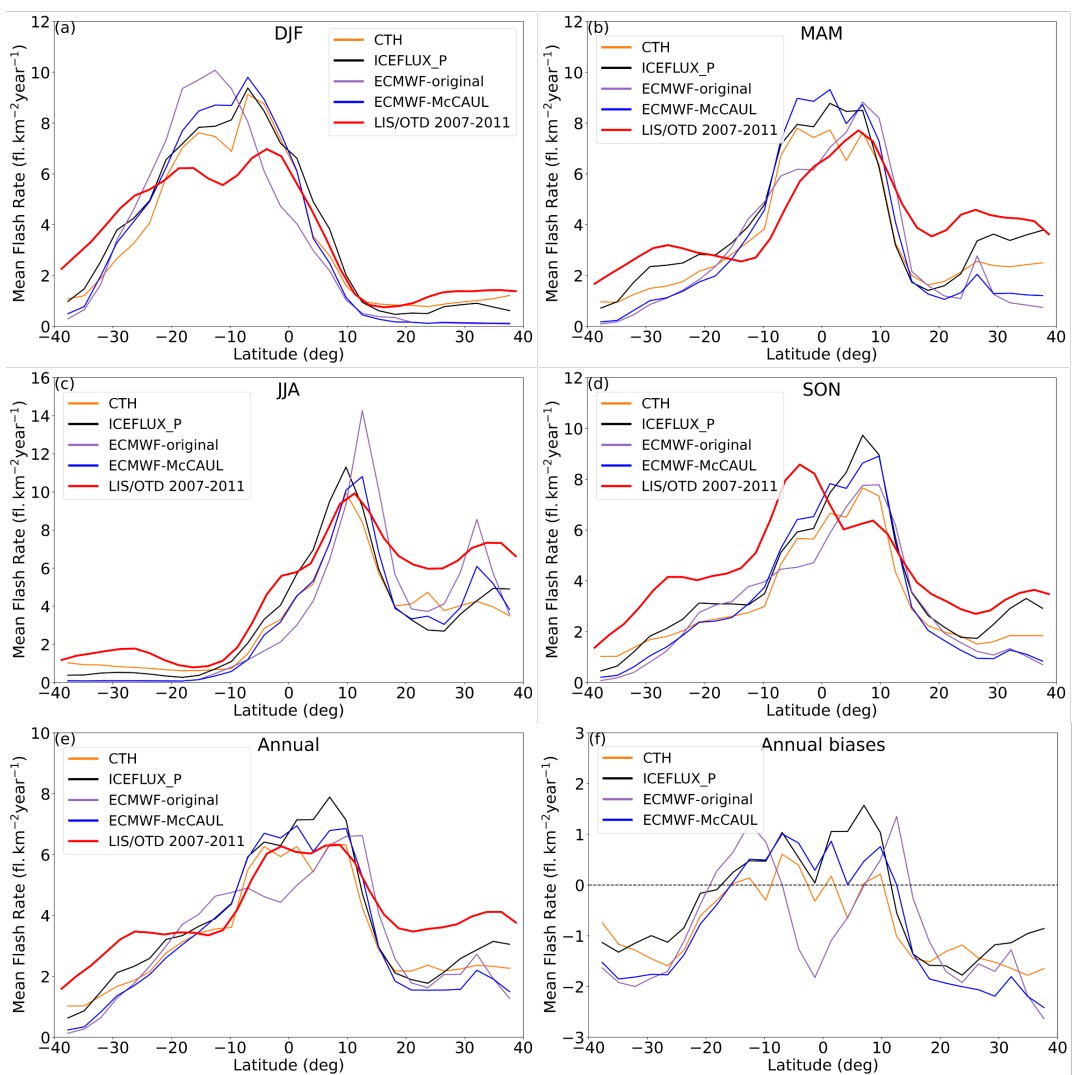

**Figure 6: Seasonal and annual zonal average lightning flash densities distribution from LIS 2007–2011 climatology (red line) and**
**from the simulation results (2007–2011) obtained using different lightning schemes. The biases (model-obs.) in Fig. 6e are also**
**presented in Fig. 6f.**

**3.2 Evaluation of LNOₓ emissions**

## 3.2 Evaluation of LNO$_x$ emissions

### 3.2.1 Evaluation of LNO$_x$ emissions by ATom1 and ATom2 observations

To evaluate the LNO$_x$ emissions of different lightning schemes, we used ATom1 and ATom2 aircraft measurements (NO) for comparison against model results. All lightning schemes, when implemented in CHASER, produce flash rates corresponding to global annual LNO$_x$ emissions within the range estimated by Schumann and Huntrieser (2007) of 2–8 TgN yr$^{-1}$. To eliminate differences in annual total LNO$_x$ emissions by different lightning schemes, we chose to adjust the annual LNO$_x$ emissions of all lightning schemes to 5.0 TgN yr$^{-1}$ by applying adjustment factors. The "backward C-shaped" LNO$_x$ vertical profile is applied to all lightning schemes.

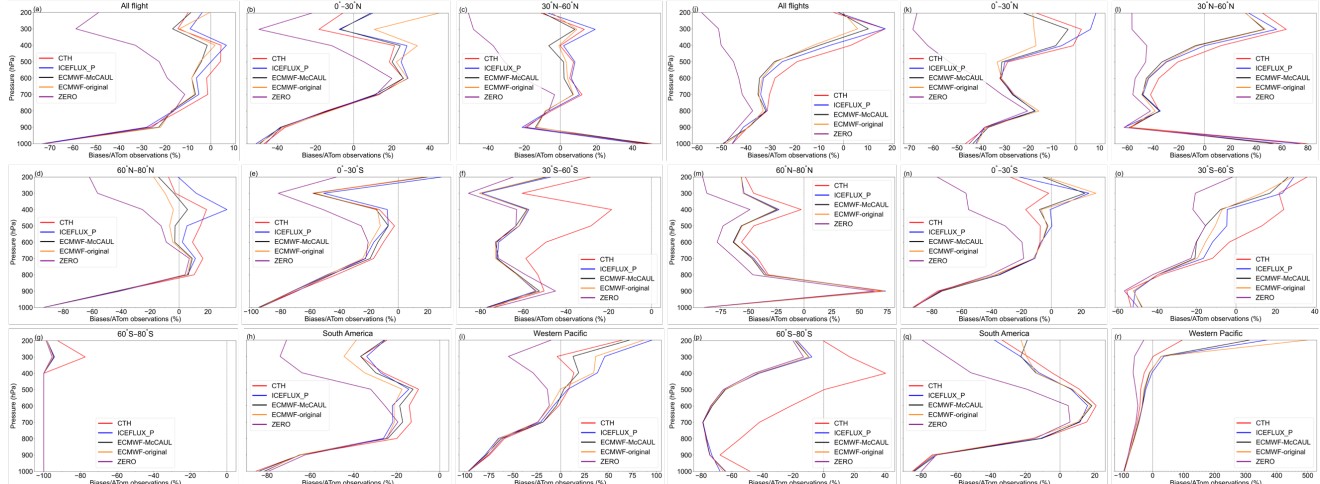

**Figure 7: Vertical profile of biases/ATom1 observations (a–i) and the vertical profile of biases/ATom2 observations (j–r). The bias is the model bias (NO) against ATom observations (NO). Data for each pressure level P are calculated within the range of P±50 hPa. South America is the region of 0°–30°S, 0°–30°W. The Western Pacific is the region of 10°N–30°S, 160°E–160°W.**

**Table 3: Model biases (NO) when compared against ATom1 (upper panel) and ATom2 (lower panel). The unit is ppt. The biases within the South America region (0–30°S, 0–30°W) and Western Pacific region (10°N–30°S, 160°E–160°W) are also shown in this table.**

| Lightning scheme | All flight | 0°–30°N | 30°N–60°N | 60°N–80°N | 0°–30°S | 30°S–60°S | 60°S–80°S | South America | Western Pacific |
|---|---|---|---|---|---|---|---|---|---|
| CTH | -6.54 | -3.22 | -0.50 | -13.06 | -9.33 | -12.32 | -7.55 | -6.79 | -3.03 |
| ICEFLUX_P | -5.18 | 0.31 | 1.15 | -9.16 | -8.21 | -16.21 | -8.28 | -7.00 | 0.08 |
| ECMWF-McCAUL | -6.99 | 0.13 | -1.05 | -14.80 | -9.43 | -16.42 | -8.29 | -7.17 | -2.24 |
| ECMWF-original | -5.48 | 7.03 | 0.28 | -16.66 | -9.59 | -16.38 | -8.30 | -8.71 | -0.72 |
| ZERO | -19.00 | -11.02 | -20.85 | -32.98 | -15.91 | -17.35 | -8.34 | -13.77 | -8.63 |
| Lightning scheme | All flight | 0°–30°N | 30°N–60°N | 60°N–80°N | 0°–30°S | 30°S–60°S | 60°S–80°S | South America | Western Pacific |
| CTH | -0.91 | -2.57 | 5.80 | -6.18 | -11.11 | 3.61 | 1.45 | -19.16 | -4.70 |
| ICEFLUX_P | -1.04 | -0.76 | 3.98 | -6.81 | -7.45 | 2.82 | -4.88 | -22.02 | 3.01 |
| ECMWF-McCAUL | -1.73 | -3.71 | 2.81 | -6.89 | -3.71 | 1.81 | -5.33 | -12.24 | 1.20 |
| ECMWF-original | -1.95 | -5.26 | 2.96 | -6.87 | -2.74 | 1.58 | -5.23 | -13.90 | 3.55 |
| ZERO | -12.66 | -15.51 | -11.08 | -9.77 | -28.40 | -4.18 | -5.94 | -47.68 | -13.14 |

Table 3 presents model biases of different lightning schemes against the ATom1 and ATom2 observations. Figure 7 displays the vertical profile of biases/ATom observations in percentage terms. In Table 3 and Fig. 7, case ZERO is the case with the lightning flash, with LNO$_x$ emissions completely switched off. Comparisons between model results and ATom observations were conducted within two specific regions (South America region and Western Pacific region) in which LNO$_x$ is the major source of NO$_x$ (Fig. 8). As Table 3 and Fig. 7 show, the model generally tends to underestimate the NO concentrations. The model biases are reduced considerably by including lightning NO$_x$ sources. For ATom1, overall, the ICEFLUX_P scheme has the smallest model bias. The original ECMWF scheme also reduced the model biases compared to the CTH scheme

(Table 3). The ICEFLUX_P and the ECMWF-McCAUL schemes reduced the model biases substantially within 0°–30°N
latitude where the lightning activities are most dominant during the ATom1 observation period (2016-07-29 ~ 2016-08-23).
In the range of 30°S to 80°N in ATom1, overall the ICEFLUX_P scheme reduced the model biases considerably and the
ECMWF schemes slightly reduced or extended the model biases compared to the CTH scheme (Table 3, Figs. 7b–e).
However, in the range of 30°S–80°S, the model biases were extended by the ICEFLUX_P and the ECMWF schemes
compared to the CTH scheme (Table 3, Figs. 7f–g).

For ATom2, overall, the ECMWF schemes slightly reduced the model biases over the upper troposphere, compared to the
CTH scheme (Fig. 7j). During the ATom2 observation period (2017-01-26 ~ 2017-02-21), the lightning activities are most
dominant within the range of 0°–30°S, where the model biases were reduced significantly by newly implemented lightning
schemes. A hotspot of lightning activities during the ATom2 observation period is the South America region, where the
model biases were reduced dramatically by the ECMWF schemes. The model biases were mostly reduced by the newly
implemented lightning schemes within the low latitude and middle latitude regions, but slightly extended within the high
latitude regions. The model biases were mostly reduced or extended over the middle to upper troposphere (Fig. 7). This is
true because most $LNO_x$ was distributed over the middle to upper troposphere. Also, $NO_x$ has a longer lifetime over the
middle to upper troposphere. In the Western Pacific region, results obtained from comparisons with ATom1 and ATom2
indicate that all lightning schemes overestimated $LNO_x$ emissions in the upper troposphere; also, both the ICEFLUX_P
scheme and ECMWF schemes reduced the total model biases considerably more than the CTH scheme did.

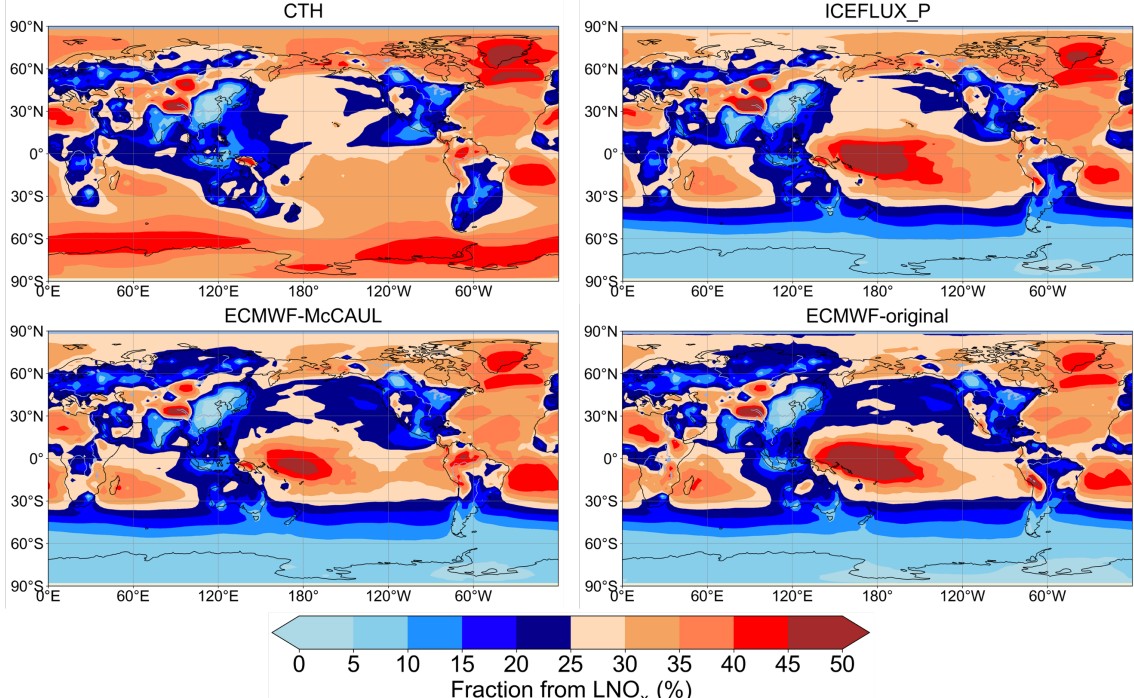

**Figure 8: Sensitivity of simulated tropospheric NO₂ columns to $LNO_x$ emissions using different lightning schemes in 2019. NO₂**
**column because of $LNO_x$ emissions was determined as the difference between the simulation with $LNO_x$ emissions and a simulation**
**that excludes $LNO_x$ emissions.**
**3.2.2 Evaluation of $LNO_x$ emissions by TROPOMI satellite observations**
TROPOMI satellite observations of tropospheric NO₂ columns were used to evaluate $LNO_x$ emission results obtained using
the CHASER model. To eliminate differences in annual total $LNO_x$ emissions attributable to the different lightning schemes,
we adjusted the annual $LNO_x$ emissions of all lightning schemes to 5.0 TgN yr⁻¹ using different adjustment factors. For direct
comparison between CHASER and TROPOMI tropospheric NO₂ columns, the averaging kernel information from
TROPOMI observations was used. The averaging kernels were applied to CHASER outputs following Eq. (16).
$X_{chaser} = \sum_{i=1}^{N} A_{tropomi} x_{chaser}$ (16)
In that equation, $X_{chaser}$ represents the CHASER tropospheric $NO_2$ column after averaging kernels applied, $A_{tropomi}$
denotes the TROPOMI averaging kernels, $x_{chaser}$ denotes the CHASER $NO_2$ partial column at layer i, and N denotes the
number of tropospheric layers.

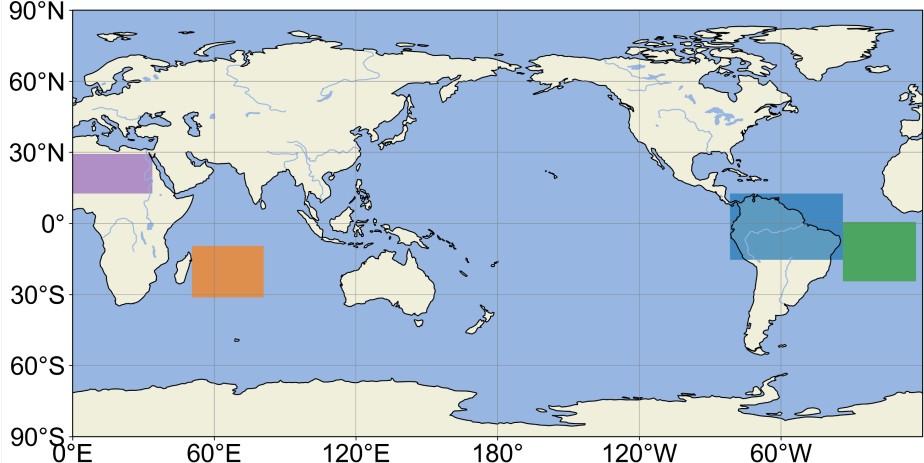

**Figure 9: Four target regions for which LNO$_x$ is the major source of NO$_x$. The four target regions are North Africa (purple),**
**Indian Ocean (orange), Amazon (blue), and South Atlantic (green).**

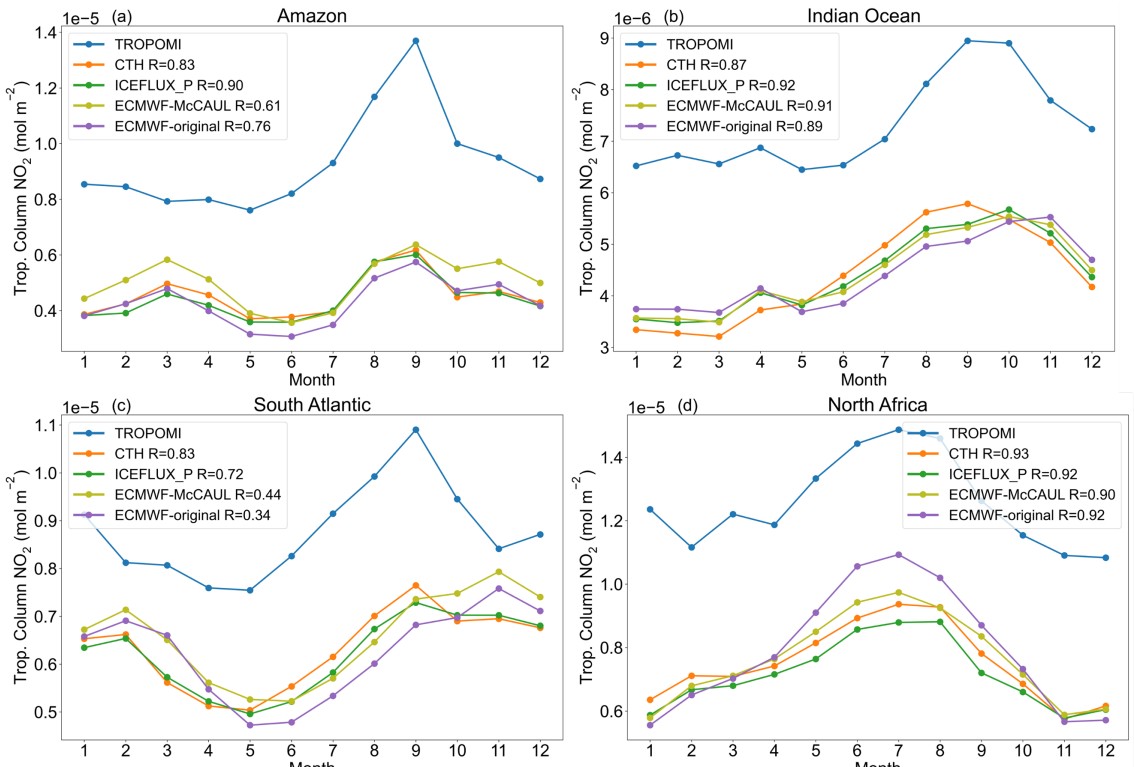

**Figure 10: Comparisons of smoothed CHASER and TROPOMI (blue) tropospheric NO$_2$ columns over four target regions in 2019.**
**Legends show the temporal correlation coefficients.**

Comparison between TROPOMI observations and CHASER outputs indicates that the CHASER model tends to
underestimate tropospheric NO$_2$ columns. Overall, the newly implemented lightning schemes have not shown improvements
of model biases of tropospheric NO2 columns at an annual global scale. To minimize the uncertainties of model biases of
tropospheric NO$_2$ columns caused by other factors, we chose to further evaluate the LNO$_x$ emissions by TROPOMI
observations over four specific regions (Fig. 9), where LNO$_x$ is the major source of NO$_x$ (as shown in Fig. 8).
Figure 10 presents a comparison of smoothed CHASER and TROPOMI tropospheric NO$_2$ columns over four target regions
in 2019. The spatial average values of each month in 2019 are shown in Fig. 10. That figure shows, generally, the model
captured the temporal variation of tropospheric NO$_2$ columns in the four regions. Actually, the temporal variations of
modelled tropospheric NO2 columns are close to each other. For the Amazon region, lightning activities are most dominant

during MAM and SON, when the ECMWF-McCAUL scheme has shown noticeable improvements in model biases (Fig. 10a). Figure 10b reveals that all the newly implemented schemes slightly reduced the model biases with the original ECMWF scheme showing the smallest model biases during the most prevailing season of lighting (DJF). Figure 10c is for the South Atlantic region where the most prevailing season of lighting is also DJF. Figure 10c shows that the ECMWF schemes slightly reduced the model biases compared to the CTH scheme. Referring to Fig. 10d, the dominant season of lightning is JJA, when the ECMWF-original scheme considerably reduced the model biases and the ECMWF-McCAUL scheme also slightly reduced the model biases.

**3.3 Effects of different lightning schemes on tropospheric chemical fields**

In the tropospheric chemical field, $LNO_x$ has an important role. The $LNO_x$ effects on the tropospheric chemical fields vary along with differences in the horizontal distribution of $LNO_x$ in different lightning schemes. To evaluate the influences of different lightning schemes on the tropospheric chemical fields, several ten-year (2011–2020) experiments were conducted with the ten-year mean $LNO_x$ production of all lightning schemes adjusted to 5.0 TgN $yr^{-1}$ (Sect. 2.4). CTH scheme with a "backward C-shaped" profile is regarded as the base scheme. The effects of different lightning schemes on the atmospheric chemistry are calculated as shown in Eq. (17).

$$Impact_{ij} = \frac{(LS_{ij} - Base_j)}{Base_j} \qquad (17)$$

Therein, $Impact_{ij}$ represents the effects of the $i$-th lightning scheme on the concentrations of target atmospheric component $j$. Also, $LS_{ij}$ denotes the concentrations of target atmospheric component $j$ simulated by the $i$-th lightning scheme. $Base_j$ stands for the concentrations of target atmospheric component $j$ as simulated using the base scheme.

Figure 11 presents the respective effects of the ECMWF-McCAUL, original ECMWF, and ICEFLUX_P schemes on the atmospheric chemical fields ($NO_x$, $O_3$, OH, CO) relative to the base scheme CTH. The ECMWF-McCAUL scheme led to an increase (approximate maximum is 12%) in $NO_x$ concentration at low latitude regions and a decrease (approximate maximum is 15%) at middle to high latitude regions. In the case of the ECMWF-McCAUL scheme, the concentration of ozone and OH radical mostly increased at low latitude regions and decreased at middle to high latitude regions in the Southern Hemisphere, which corresponds to the changing pattern of $NO_x$. The effects of the original ECMWF scheme on the atmospheric chemical fields are similar to that of the ECMWF-McCAUL scheme. However, the original ECMWF scheme led to a higher total tropospheric CO burden compared to the ECMWF-McCAUL scheme. As Fig. 11 shows, the three lightning schemes led to a marked decrease in $NO_x$, $O_3$, and OH radical concentrations over the South Pole region. This decrease occurred because the lightning densities and the $LNO_x$ emissions simulated by the CTH scheme are markedly higher than those simulated using other lightning schemes at this latitude band (Fig. 2). Moreover, $NO_x$ can engender the formation of ozone and OH radical. In the case of the ICEFLUX_P scheme, the concentrations of $NO_x$, ozone, and OH radical mostly increased in the tropics and decreased at middle to high latitude regions in the Southern Hemisphere.

Methane lifetime is an indicator reflecting the tropospheric oxidation capacity. The global mean tropospheric lifetime of methane against tropospheric OH radical spanning 2011–2020 with the CTH, original ECMWF, ECMWF-McCAUL, and ICEFLUX_P schemes are estimated respectively as 9.226 years, 9.299 years, 9.256 years, and 9.229 years. Compared to the CTH scheme, the ECMWF schemes led to a slight increase in methane's global mean tropospheric lifetime. In contrast, the methane's global mean tropospheric lifetime simulated by the ICEFLUX_P scheme is almost the same as that simulated by the CTH scheme. Although little difference exists in the total tropospheric oxidation capacity simulated by different lightning schemes, the ECMWF schemes and ICEFLUX_P scheme led to marked changes of oxidation capacity in different regions of the troposphere.

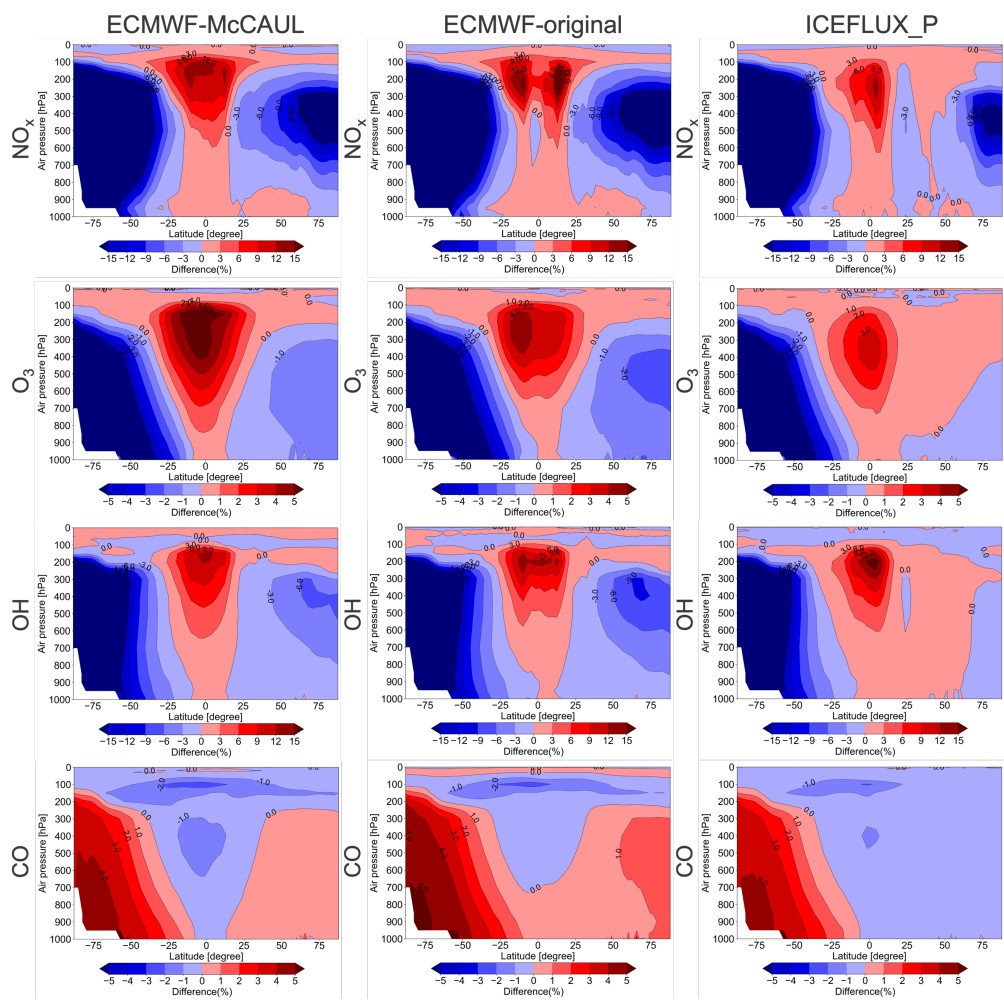

**Figure 11: Effects of ECMWF-McCAUL scheme, original ECMWF scheme, and ICEFLUX_P scheme on the atmospheric chemical**
**fields (NOₓ, O₃, OH, CO) relative to the CTH scheme on the zonal mean (%).**

## 3.4 Historical trend analysis of lightning density

The accuracy of predicting the simulated lightning distribution under the current climate is only one aspect of lightning
scheme evaluation. The ability of the lightning scheme to reproduce the trend of lightning under climate change is also an
important factor. For this study, 20 years of (2001–2020) experiments were conducted to analyze the historical trends of
lightning flash rates simulated using different lightning schemes (Sect. 2.4).

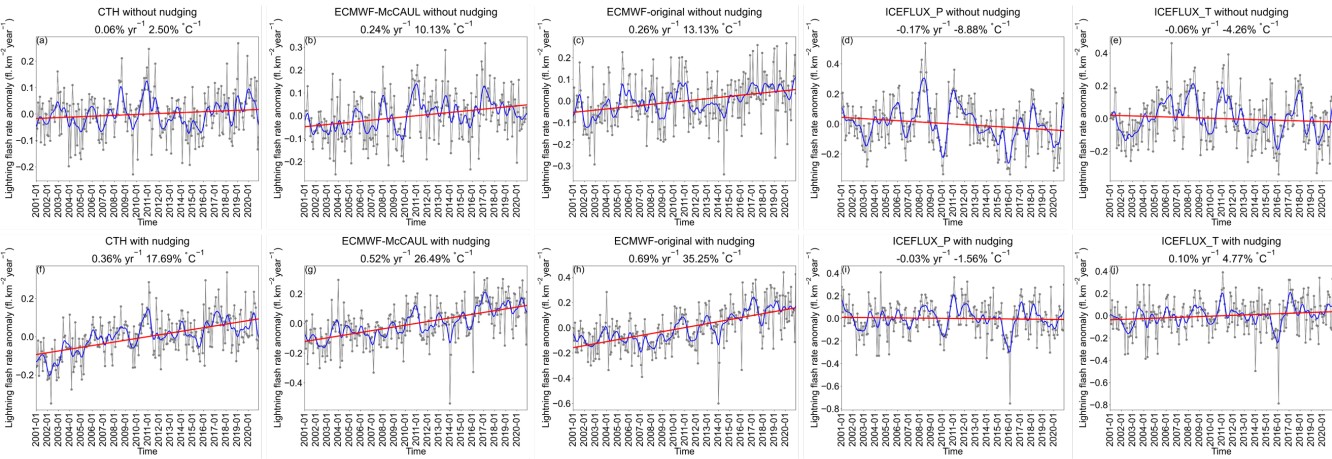

**Figure 12: Global anomaly of lightning flash rates calculated from simulation results (2001–2020) using different lightning schemes.**
**Figures 12(a–e) present results without nudging; Figs. 12(f–j) present results with nudging. The grey lines with points represent the**
**monthly time-series data of the global mean lightning flash rate anomaly. The blue curves represent the monthly time-series data of**
**the global mean lightning flash rate anomaly with the 1-D Gaussian (Denoising) Filter applied. The red lines are the fitting curves**
**of the grey lines.**

**Table 4: Changes in global mean surface temperature (ΔTS), global mean lightning flash rate (ΔLFR), and the rate of change of lightning flash rate corresponding to each degree Celsius increase in global mean surface temperature (ΔLFR/ΔTS). The upper panel shows results obtained without nudging. The lower panel presents results obtained with nudging. Changes were obtained by calculating the difference between the rightmost and leftmost points of the approximating curve for the 2001–2020 time-series data.**

| Lightning scheme | ΔTS (°C) | ΔLFR (%) | ΔLFR/ΔTS (%/°C) |
|---|---|---|---|
| CTH | 0.38 | 0.95 | 2.50 |
| ECMWF-McCAUL | 0.39 | 3.95 | 10.13 |
| ECMWF-original | 0.40 | 5.25 | 13.13 |
| ICEFLUX_P | 0.40 | -3.55 | -8.88 |
| ICEFLUX_T | 0.34 | -1.45 | -4.26 |
| Lightning scheme | ΔTS (°C) | ΔLFR (%) | ΔLFR/ΔTS (%/°C) |
| CTH | 0.39 | 6.90 | 17.69 |
| ECMWF-McCAUL | 0.39 | 10.33 | 26.49 |
| ECMWF-original | 0.39 | 13.74 | 35.23 |
| ICEFLUX_P | 0.39 | -0.61 | -1.56 |
| ICEFLUX_T | 0.39 | 1.86 | 4.77 |

Figure 12 shows the global anomaly of lightning flash rates calculated from the simulation results. Because nudging to meteorological reanalysis data cannot be used when predicting lightning trends under future climate changes, we also showed the results without nudging. The un-nudged runs also represented the short-term surface warming like the experiments with nudging. The only differences between the un-nudged and nudged experiments are whether the meteorological fields are nudged towards the six-hourly NCEP FNL data. We used the Mann–Kendall rank statistic to ascertain whether the lightning trends in Fig. 12 are significant (Hussain et al., 2019). From results of the Mann–Kendall rank statistic test (significance set as 5%), all the trends in Fig. 12 were inferred as significant except for the trends shown in Figs. 12a, e, i. As Fig. 12 shows, all lightning schemes predicted increasing trends or no significant trends of lightning except the ICEFLUX_P scheme without nudging, which predicted a decreasing lightning trend. The isotherms alternative application of ICEFLUX (ICEFLUX_T) led to slightly enhanced lightning trends toward positive lighting trends compared to the ICEFLUX_P scheme. As explained by Romps (2019), the ICEFLUX_P approach is based on a fixed isobar which makes it less convenient for climate change studies. Therefore, at least the lightning trends simulated by the ICEFLUX_T approach are expected closer to the real situation than the ICEFLUX_P approach.

As displayed in Fig. 12, the positive lightning trends are generally enhanced by application of meteorological nudging. A further investigation of the trends of CAPE during 2001–2020 discloses that the trends of global averaged CAPE are also enhanced by application of meteorological nudging. Since higher CAPE means higher buoyancy in the updrafts, which led to the higher lightning densities calculated by the lightning schemes used in this study. It is worth noting that even though the meteorological fields (u, v, T) of nudged simulations are expected closer to the real situations, we cannot analogously deduce that the lightning trends predicted by the nudged runs are also closer to the real situations. This is because the predicted lightning trends are not only controlled by the meteorological fields but also controlled by many other physical processes (e.g., cumulus parameterization).

Few studies have specifically examined the lightning trends predicted by the ECMWF schemes under the short-term surface warming. When nudging was not applied, the ECMWF schemes predicted the increasing trends of lightning flash rates under

the short-term surface warming by factors of 4 (ECMWF-McCAUL scheme) and 5 (original ECMWF scheme) compared to
the CTH scheme (Table 4).

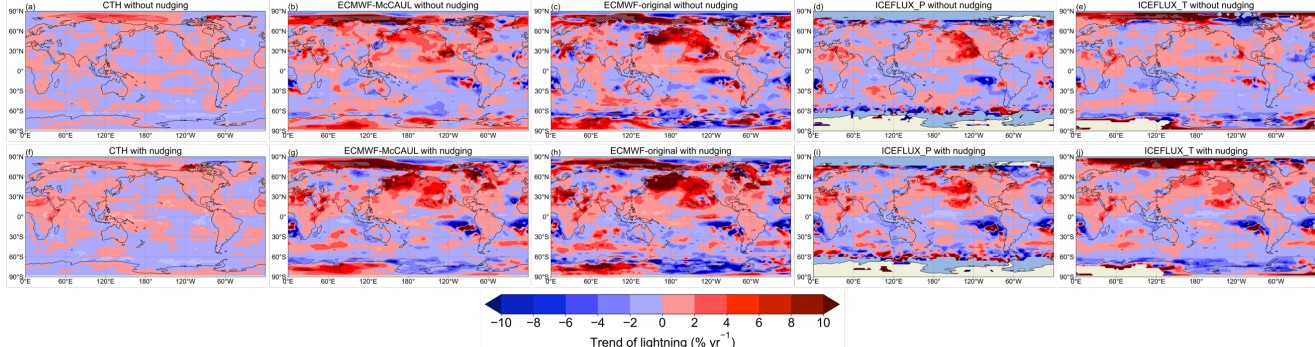

**Figure 13: Changes in the lightning flash rate (% yr⁻¹) during 2001–2020 on the two-dimensional map. Changes at every point were**
**calculated from the function of approximating curve for the 2001–2020 time-series data at each grid cell. Figures 13(a–e) show results**
**without nudging; Figs. 13(f–j) show results with nudging. There are some missing values in the case of the ICEFLUX scheme because**
**the upward cloud ice flux used is diagnosed as zero by the CHASER model typically within the high latitude regions.**

Figure 13 shows a global map of changes in the lightning flash rate (% yr⁻¹) during 2001–2020. In Fig. 13, the area in which
the trend was found to be significant by the Mann–Kendall rank statistic test (significance level inferred for 5%) is marked
with hatched lines. As Fig. 13 shows, the distribution of trends simulated by the same lightning scheme is similar whether
meteorological nudging was applied or not. As displayed in Fig. 13, in the Arctic region of the Eastern Hemisphere, both the
CTH scheme and the ECMWF schemes showed an increasing trend of lightning. Earlier studies based on the World Wide
Lightning Location Network (WWLLN) lightning observations have indicated that lightning densities in the Arctic increase
concomitantly with increasing global mean surface temperature (Holzworth et al., 2021). Earlier studies indicate that the
results of the CTH scheme and the ECMWF schemes are reasonable for the Arctic region of the Eastern Hemisphere. In the
high latitude region of the Southern Hemisphere (60°S–70°S), both the CTH scheme and the ECMWF schemes showed
decreasing lightning trends. Lightning is rarely observed south of 60°S (Kelley et al., 2018). Moreover, the trends of
lightning in this region expected to occur with the short-term surface warming remain highly uncertain. In some parts of the
Northern Pacific Ocean, the ECMWF schemes and ICEFLUX scheme results showed increasing trends of lightning, which is
consistent with results obtained from an earlier study (Walter and Buechler, 2008). All schemes show decreasing trends for
lightning flash rates in Indonesia, although only the ICEFLUX scheme explicitly passed the significance test. In the North
Atlantic, all schemes showed increasing lightning trends. Only the CTH scheme failed the significance test.
**3.5 Effects of LNOₓ emissions on trends of tropospheric O₃ and NOₓ columns**
The historical trends of lightning densities during 2001–2020 calculated using different lightning schemes have been
discussed in Sect. 3.4. Increasing or decreasing trends of lightning can engender corresponding trends of LNOₓ emissions,
which can consequently influence trends of NOₓ and O₃ concentrations. To ascertain the extent to which the LNOₓ emissions
influence NOₓ and O₃ concentration trends, the effects of the LNOₓ emissions on the trends of tropospheric NOₓ and O₃
columns have been estimated and discussed. We conducted two sets of experiments (Sect. 2.4), one of which interactively
calculated LNOₓ emission rates, whereas the other one maintained the 2001 LNOₓ emission rates for simulations of the
entire 20 years. The LNOₓ emission effects on the trends of tropospheric NOₓ and O₃ columns can be estimated
quantitatively by comparing the results of these two sets of experiments. We also conducted the verification of the simulated
trends of tropospheric NOₓ and O₃ columns by the OMI satellite observations and the results are exhibited in Fig. S1 and Fig.
S2. Generally, the model has well captured the trends of global averaged tropospheric NO₂ and O₃ columns even though the
trends of both tropospheric NO₂ and O₃ columns are underestimated by the model. Overall, it is obvious that the modelled
trends with interannually varying LNOₓ emissions with nudging are most close to the OMI observations.

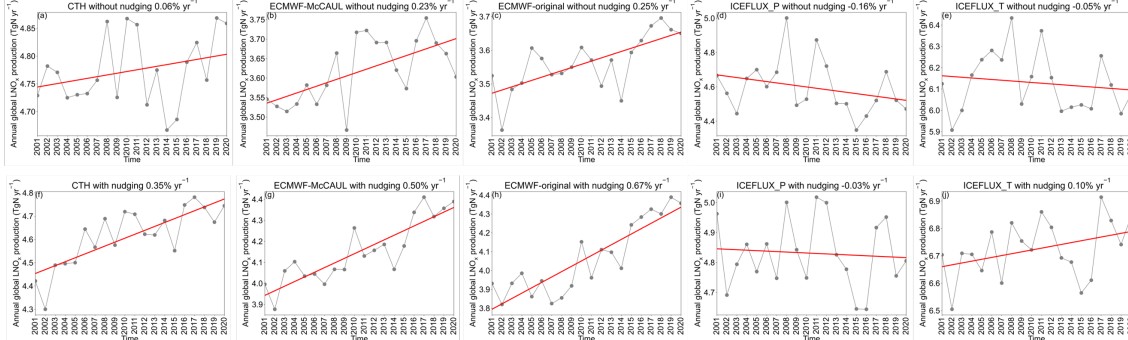

**Figure 14: Trends of annual global LNO$_x$ emissions calculated from simulation results (2001–2020) from different lightning schemes.**
**Red lines are fitting curves. Figures 14(a–e) present results without nudging; Figs. 14(f–j) present results with nudging. The number**
**in the title of each figure represents the trend corresponding to that figure in the unit of % yr$^{-1}$.**

Figure 14 presents trends of annual global LNO$_x$ emissions calculated from the simulation results (2001–2020) obtained
using different lightning schemes. As Fig. 14 shows, the annual global LNO$_x$ emission trends correspond to the trends of
lightning presented in Fig. 12. Similar to the trends found for lightning, the trends of annual global LNO$_x$ emissions are also
increased by application of meteorological nudging. Only the ICEFLUX scheme simulated decreasing trends of annual
global LNO$_x$ emissions.

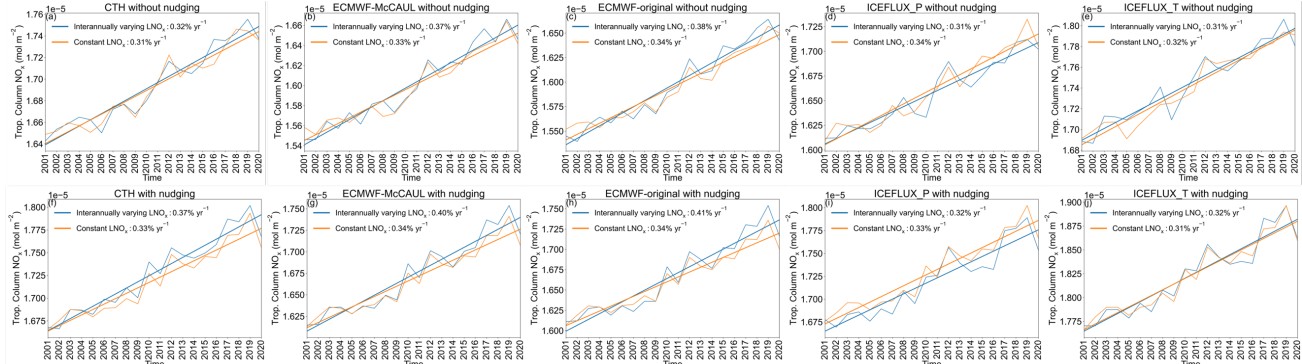

**Figure 15: Trends of global mean tropospheric NO$_x$ columns calculated from simulation results (2001–2020) using different lightning**
**schemes. Straight lines in the figure are the fitting curves. The numbers in legends represent trends corresponding to that figure in**
**the unit of % yr$^{-1}$. Figures 15(a–e) present results obtained without nudging; Figs. 15(f–j) present results obtained with nudging.**

Figure 15 portrays trends of global mean tropospheric NO$_x$ columns calculated from the first and second set of experiments
(Table 2). As Fig. 14 and Fig. 15 depict, when the trends of annual global LNO$_x$ emissions are not strong (e.g., Fig. 14a),
their effects on the trends of global mean tropospheric NO$_x$ columns are negligible. The marked increasing trends of annual
global LNO$_x$ emissions (Figs. 14f, g, h) led to great increases (12.12%–20.59%) of the increasing trends of tropospheric NO$_x$
columns (Figs. 15f, g, h). In the case of the ICEFLUX_P scheme without nudging, because of the decreasing trends of LNO$_x$
emissions, the increasing trends of the tropospheric NO$_x$ columns decreased by around 10%.

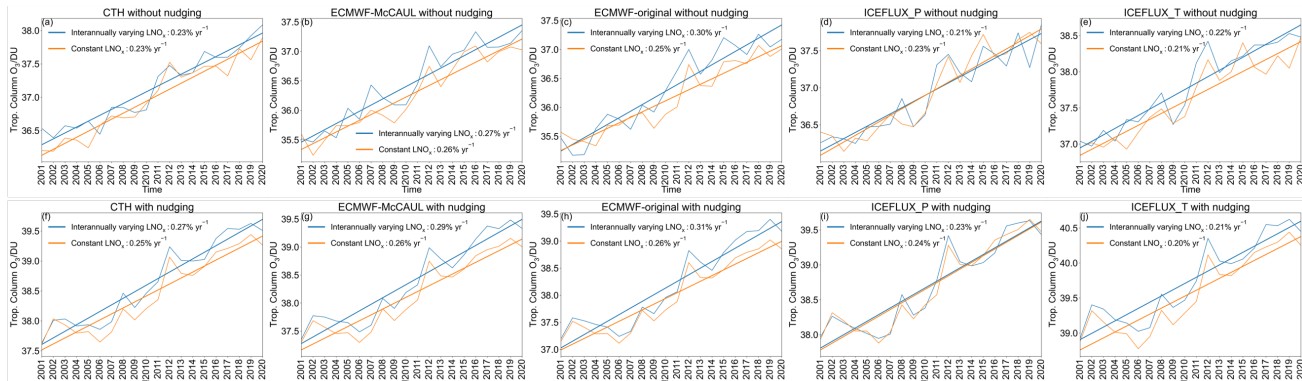

**Figure 16: Trends of global mean tropospheric O$_3$ columns calculated from simulation results (2001–2020) using different lightning**
**schemes. Straight lines in the figure are the fitting curves. The number in the legend represents the trend corresponding to that**
**figure in the unit of % yr$^{-1}$. Figures 16(a–e) present results obtained without nudging; Figs. 16(f–j) show results obtained with**
**nudging.**

Figure 16 is similar to the results shown in Fig. 15, but for tropospheric $O_3$ columns. Because $NO_x$ causes the formation of
$O_3$ by the fundamental chemical cycle of $O_x$–$NO_x$–$HO_x$, the trends of the global mean tropospheric $O_3$ columns are affected
strongly by trends of the global mean tropospheric $NO_x$ columns. In some cases, the simulated trends of tropospheric $O_3$
columns are almost identical as portrayed in Figs. 16 a, b, e, i, j because the trends of tropospheric $NO_x$ columns simulated
by the two sets of experiments are very similar (Figs. 15 a, b, e, i, j). As Fig. 14 and Fig. 16 show, the marked increasing
trends of annual global $LNO_x$ emissions led to increases of the increasing trends of tropospheric $O_3$ columns by around 15%
(Figs. 16f, g, h). In the case of ICEFLUX_P without nudging, because of the decreasing trend of $LNO_x$ emissions, the
increasing trend of the tropospheric $O_3$ columns decreased by around 10% (Fig. 16d). Note that the tropospheric $NO_x$ or $O_3$
columns in 2001 simulated by the first set of experiments and the second set of experiments are not exactly the same. This is
because the blue lines show results with interactively calculated $LNO_x$ emission rates (the time resolution is 10–30 minutes).
But the orange lines show results calculated by reading daily mean input data for $LNO_x$ emission rates, which inhibits
interaction of $LNO_x$ with meteorology in the model.

In conclusion, because the ICEFLUX scheme predicts the opposite trends of $LNO_x$ emissions from the other lightning
schemes, they simulate opposite effects on the historical trends of global mean tropospheric $NO_x$ and $O_3$ columns.
Furthermore, an evident trend of annual global $LNO_x$ emissions has a strong effect on the trend of global mean tropospheric
$NO_x$ and $O_3$ columns.
**4 Discussions and Conclusions**
Three new lightning schemes, the ICEFLUX, the original ECMWF, and the ECMWF-McCAUL schemes were implemented
into CHASER (MIROC), a global chemical climate model. Using LIS/OTD lightning observations as validation data, both
the ICEFLUX_P and ECMWF schemes simulated the spatial distribution of lightning more accurately on a global scale than
the CTH scheme did, and the lightning distribution in the ocean region was especially improved. The ECMWF-McCAUL
scheme showed the highest prediction accuracy for the spatial distribution of lightning on a global scale. It is noteworthy that
whilst the ice-based parametrisations showed superb prediction accuracy of lightning distribution under today's climate, they
have greater uncertainties associated with inputs, especially regarding the microphysics scheme used (Charn and Parishani,

636 2021).


To verify the $LNO_x$ emissions of different lightning schemes, we used NO observations from ATom1 and ATom2. Overall,
both the ICEFLUX_P scheme and the ECMWF schemes partially reduced model biases typically over the dominant regions
of lightning activities compared to the CTH scheme. We also used TROPOMI tropospheric $NO_2$ columns to verify the $LNO_x$
emissions of different lightning schemes. Although the ICEFLUX_P and the ECMWF schemes have not shown
improvements of model biases of tropospheric $NO_2$ columns at an annual global scale, they generally led to an obvious
reduction of model biases in the prevailing seasons of lightning within the regions where $LNO_x$ is a dominant source of $NO_x$.
Several studies have pointed out that the TROPOMI data used in this study biased negatively compared to the airborne or
ground-based observation data (Tack et al., 2021; Verhoelst et al., 2021; van Geffen et al., 2022). Since the TROPOMI data
used are generally negatively biased and the simulated tropospheric $NO_2$ columns are underestimated compared to the
TROPOMI observations. Therefore, the uncertainties that existed in the TROPOMI data have negligible impacts on the
conclusions of our study.

Effects of the newly implemented lightning schemes on the tropospheric chemical fields are evaluated compared to the CTH
scheme. Compared with the CTH scheme, the ECMWF schemes mainly led to a slight increase in $NO_x$, ozone, and OH
radical concentrations at low latitude regions and a decrease at middle-latitude to high-latitude regions. Effects of the
ICEFLUX_P scheme on the tropospheric chemical fields slightly differ from those of the ECMWF schemes. The
ICEFLUX_P scheme mainly causes a slight increase of $NO_x$, ozone, and OH radical concentrations from the tropics to the
Northern Hemisphere and a decrease in the concentrations of the three chemical species in the Southern Hemisphere except
the tropics. The commonality between the ECMWF schemes and the ICEFLUX_P scheme is that they both result in
decreasing concentrations of $NO_x$, ozone, and OH radical at the middle to high latitude regions of the Southern Hemisphere.
Although the newly implemented lightning schemes have little effect on the total oxidation capacity of the troposphere
compared to the CTH scheme, they led to marked changes of oxidation capacity in different regions of the atmosphere.

This study also analyzed the historical trends of lightning simulated by different lightning schemes under the short-term
surface warming during 2001–2020. The Mann–Kendall rank statistic was used to ascertain whether the lightning trends
were significant. Use of Mann–Kendall rank statistic tests revealed that all the simulated historical lightning trends are
significant, except the CTH and the ICEFLUX_T schemes without nudging and the ICEFLUX_P scheme with nudging, for
significance at 5%. All the lightning schemes predicted increasing lightning trends or no significant trends except the
ICEFLUX_P scheme without nudging, which predicted a decreasing lightning trend. The ICEFLUX_T scheme predicted a
decreasing trend without nudging even though the trend failed the significant test. If it's accepted that the non-inductive
charging mechanism is an appropriate basis for a lightning parametrisation, then the implication is that in the future if cloud
ice (and cloud ice fluxes) reduce then electrical charging will reduce too. This provides a line of scientific reasoning to
explain why lightning may reduce in the future. Moreover, findings showed that when nudging was not applied, the
ECMWF schemes predicted an increasing trend of lightning flash rate under the short-term surface warming by factors of 4
(ECMWF-McCAUL scheme) and 5 (original ECMWF scheme) compared to the CTH scheme. Although a considerable
degree of uncertainty remains in determining the sensitivity of lightning activity to changes in surface temperature on the
decadal timescale (Williams 2005), the majority of past estimates show the sensitivity tends average close to 10% $K^{-1}$ (Betz
et al., 2008, p. 521). This value is most consistent with the lightning increase rate predicted by the ECMWF-McCAUL
scheme without nudging in this study. Future research should be undertaken for specific examination of development of
lightning schemes that both accurately predict the global distribution of $LNO_x$, and which predict the changes in lightning
that are expected to occur concomitantly with global climate change. Finally, we quantitatively estimated the $LNO_x$ emission
effects on tropospheric $NO_x$ and $O_3$ column trends during 2001–2020. Results showed that a marked trend of annual global
$LNO_x$ emissions significantly affects the trend of global mean tropospheric $NO_x$ and $O_3$ columns.
**Code availability**
The source code for CHASER to reproduce results in this work is obtainable from the repository at
https://doi.org/10.5281/zenodo.5835796 (He et al., 2022)
**Data availability**
The LIS/OTD data used for this study are available from https://ghrc.nsstc.nasa.gov/hydro/?q=LRTS (last access: 11 January
2022). The ATom data used for this study are available from https://daac.ornl.gov/ATOM/guides/ATom_merge.html (last
access: 11 January 2022). The TROPOMI data used for this study are available from
https://s5phub.copernicus.eu/dhus/#/home (last access: 11 January 2022). The OMI level-3 daily global gridded (0.25° ×
0.25°) Nitrogen Dioxide product (OMNO2d) used for this study is available from
https://disc.gsfc.nasa.gov/datasets/OMNO2d_003/summary (last access: 25 May 2022). The OMI/MSL tropospheric column
ozone data used for this study are available from https://acd-ext.gsfc.nasa.gov/Data_services/cloud_slice/new_data.html (last
access: 25 May 2022).

**Author contribution**

YFH introduced new lightning schemes into CHASER (MIROC) by adding new codes to CHASER (MIROC), conducted all simulations, interpreted the results, and wrote the manuscript. KS developed the model code, conceived of the presented idea, and supervised the findings of this work and the manuscript preparation. HMSH provided the TROPOMI data and the relevant codes to pre-process the TROPOMI data.

**Competing interests**

The authors declare that they have no conflict of interest.

**Acknowledgements**

This research was supported by the Global Environment Research Fund (S–12 and S–20) of the Ministry of the Environment (MOE), Japan, and JSPS KAKENHI Grant Numbers: JP20H04320, JP19H05669, and JP19H04235. This work was supported by Japan Science and Technology Agency (JST) Support for Pioneering Research Initiated by the Next Generation (SPRAING), Grant Number JPMJSP2125. The author (Initial) would like to take this opportunity to thank the "Interdisciplinary Frontier Next-Generation Researcher Program of the Tokai Higher Education and Research System." The simulations were completed with the supercomputer (NEC SX-Aurora TSUBASA) at NIES (Japan). Thanks to NASA scientists and staff for providing LIS/OTD lightning observation data (https://ghrc.nsstc.nasa.gov/uso/ds_docs/lis_climatology/LISOTD_climatology_dataset.html, last access: 9 January 2022), ATom data (https://espo.nasa.gov/atom/content/ATom, last access: 9 January 2022), and OMI satellite observation data (https://disc.gsfc.nasa.gov/datasets/OMNO2d_003/summary, last access: 25 May 2022; https://acd-ext.gsfc.nasa.gov/Data_services/cloud_slice/new_data.html, last access: 25 May 2022). We are grateful to ESA scientists and staff for providing TROPOMI data (http://www.tropomi.eu, last access: 9 January 2022). We thank Yannick Copin for software he developed to help us with the Taylor diagram.

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
