# Peer review of "Introducing new lightning schemes into the CHASER (MIROC) chemistry climate model"

_Geoscientific Model Development, 2022_

## Author Comment (AC1)

**Response to all Anonymous Referees' comments,**

We thank the Anonymous Referees for our manuscript's thorough and constructive comments. These comments are very helpful to improve the quality of the manuscript. We have carefully revised our manuscript based on the Referees' comments. Now I respond to the Referees' comments point by point and explicitly point out the changes with line numbers in the revised manuscript. Note that the line number (or Section or Figure number etc.) in blue is the line number (or Section or Figure number etc.) in the preprint and the line number (or Section or Figure number etc.) in orange is the line number (or Section or Figure number etc.) in our revised manuscript.

**Referee#1's comment:**

This study uses the CHASER chemistry model, along with a number of observational products of lightning and atmospheric chemicals from aircraft and satellites. A number of lightning parametrisations were implemented and compared. The study concludes that several lightning schemes reduced certain biases when compared to the often-used, cloud-top height lightning scheme. 20-year trends were also compared, both with and without nudging to reanalysis meteorology. The show a wide range of trends from negative trends to some that are much higher than found with the cloud-top height lightning scheme.

Overall the paper is well written. It provides a useful demonstration of these different lightning schemes, and explores to a high level of detail how they interact with atmospheric chemistry. Notably, considering how trends in lightning $NO_x$ emissions impact trends in column $NO_x$ and ozone. There are a few general points that I think need addressed before it is suitable for publication.

**Author comment:** We sincerely thank your dedicated time and the overall positive assessment of the manuscript.

**Referee#1's comment:** The lightning observations compared to are not for the same time period. I appreciate this is to try and look at near-global observations, but I propose below suitable sensitivity analysis to assess the uncertainty introduced by the inconsistency in time period.

L245. Whilst I think it is of interest to have compared to OTD, it is a completely different 5 year period. It is quite possible that interdecadal/annual variability would influence results. I strongly recommend checking this by reproducing results within +-45degrees. For this domain you can compare against LIS for the same years, and then against OTD just in that domain. The difference in the statistics compared against OTD, opposed to LIS, will show you how much error may be in your +-75 degree statistics. Alternatively, is there not a full LIS/OTD climatology product, which would mean at least you were using a good number of years (>10years) for the tropics climatology.

**Author comment:** We agree with the referee that there may exist errors when comparing model results with different 5-year period OTD observations. In accordance with the referee's advice, we have reproduced the results by comparing the model results with the LIS observations within the same period (2007-2011) and ±38 degrees (Figure3). We also compared the 2007-2011 model results with the 1996-2000 LIS/OTD observations within ±38 degrees (The results are not shown in the revised manuscript). We found the differences in the statistics compared against 1996-2000 LIS/OTD, opposed to 2007-2011 LIS are relatively small and the conclusions are almost the same. Furthermore, we have compared the 1996-2000 LIS/OTD with the 2007-2011 LIS within ±38 degrees. We found that the spatial correlation coefficient between 2007-2011 LIS and 1996-2000 LIS/OTD is very high (R=0.99) and the relative bias is relatively small (0.65%). Based on the above analysis, although relatively small errors will occur due to the time discrepancy between the comparisons, it is considered to be a reasonable and meaningful comparison and the conclusions are reliable. We have added the above discussion into the manuscript (L305~L309). For the use of the LIS/OTD data longer than 10 years for the tropics lightning climatologies. We have tried to

evaluate the simulated lightning climatologies with the LIS/OTD 2001-2011 climatologies. We found that there are no significant differences in the statistics whether we use 5-years (2007-2011) or 11-years (2001-2011) LIS/OTD climatologies to evaluate the model. Also as suggested by Finney et al. (2014), 5 years data are necessary and appropriate to produce a lightning climatology.

**Referee#1's comment:** Whilst differences in correlations/stats can be, and are, described, you should acknowledge where these differences are sufficiently small that they may not be real (e.g. correlations within a few 0.01s of each other). Assess whether it is appropriate to draw conclusions from differences of this magnitude, especially given uncertainties in observations (even if you were using the same time period).

**Author comment:** We agree. Hence, we have conducted a significant test by considering the uncertainties in the LIS/OTD observations. This part is described in detail in the revised manuscript (L339~L355).

**Referee#1's comment:** The modelled results are actually fairly similar in many cases, as such it should be acknowledged that the input variables for some (e.g. cloud-top height) might be better observed and with a better understood response to climate change, than others (e.g. cloud ice). As this may be a more important factor for some modellers, than the small differences in bias found here.

**Author comment:** We acknowledge that the input variables for some (e.g. cloud-top height) might be better observed and with a better-understood response to climate change than others (e.g. cloud ice). We also note that the cloud-top height scheme is not considered ideal because it lacks a direct, physical link with the charging mechanism (Finney et al., 2014). Moreover, the cloud-top height scheme has a fifth-power relationship for land which leads to large errors for any model biases in cloud-top height. Therefore, it is still valuable to implement and estimate new lightning schemes which have a direct physical link with the charging mechanism such as ICEFLUX and ECMWF schemes.

**Referee#1's comment:**

Specific major comments

L23/section3.4. I don't agree that you can call this "long-term" trend analysis. 20 years is considered to be on the scale of internal variability of climate, and not even as long as the normal averaging period for looking at trends, let alone looking at the trend within that time period. I think it is helpful to show the trends but I would be more refrained in how you describe them. they may indicate the response to warming but cannot be assumed to be the same as the climate response. I would avoid saying "under global warming" and instead say it could be a "short-term response to surface warming" since if there is anything other than decadal variability in the signal, then it is the transient climate response not the equilibrium response. You should note this if you are comparing to other estimates, because they may be looking at the equilibrium response.

**Author comment:** We acknowledge that the word "long-term" here is inappropriate and we have changed "long-term trend analysis" to "historical trend analysis" in the manuscript. We also revised "under global warming" to "under the short-term surface warming" as you suggested.

**Referee#1's comment:**

L128. I'm not sure where is best to acknowledge it, but I think you need to highlight the issues around using a constant isobar for climate change simulations. As much as it is not an issue for your simulation necessarily, you are trying to relate your results to climate change and if there is significant climate change the same isobar may not be appropriate. In Finney (2018) a different isobar was used for 2100 RCP8.5. A potentially neater solution is to use an isotherm as proposed by Romps (2019).

If you have tried this it would be good to include what you found, but if not I think it would be helpful to at least acknowledge the isotherm-alternative as a more flexible approach to applying the ICEFLUX parametrisation.

**Author comment:**

We acknowledge that the isotherm-alternative is a more flexible approach to applying the ICEFLUX parametrization typically for climate change simulations. We have conducted further experiments which used the isotherm-alternative approach and added the results and discussions to our manuscript (L153~L158; Sect. 3.4 and Sect. 3.5; newly added supplement Fig. S1 and Fig. S2).

**Referee#1's comment:**

Fig14. If both green and red lines use the same $LNO_x$ emissions in 2001, how can they have different column $NO_x$ in 2001? What is the differences between the models? Spinup? This is particularly concerning in panel h. You need explain these lines begin, and even end up offset, whilst the other panels don't.

**Author comment:**

There are two reasons for this.

1. The tropospheric $NO_x$ column is different in 2001 because the red line shows results with interactively calculated $LNO_x$ emission rates (the time resolution is 10-30 minutes). But the green line shows results calculated by reading daily mean input data for $LNO_x$ emission rates, which inhibits interaction of $LNO_x$ with meteorology in the model.

2. The curves displayed in Fig 14 in our preprint applied a 1-D Gaussian Filter which exaggerated the difference in 2001.

We have solved this problem by removing the 1-D Gaussian Filter. We also have updated the figures in our revised manuscript (Figure 15 and Figure 16). Furthermore, the above first reason is also included in our revised manuscript (L618~L622).

**Referee#1's comment:**

North Pacific analysis. I can't see that this is particularly interesting. I would remove. Whilst there are some notable trends in the north pacific, the absolute amount of $LNO_x$ emission is small so it's unlikely to have a major effect on column $NO_x$ and $O_3$ in this region. As indeed you have found. Your own fig7 shows that none of the schemes result in a big effect from lightning on column $NO_2$ in the north pacific.

**Author comment:**

We have removed the North Pacific analysis from the manuscript as you suggested.

**Referee#1's comment:**

597. I don't think you can use a review from 2008 (or 2009 in the bibliography!) to say what recent estimates say are expected. Recent estimates (within the last decade of studies) are highly conflicting, and it's hard to say that any particular trend can be expected. You could say the majority of estimates show a positive trend, and tend average close to 10%/degree, but I would struggle to see a justification for anything stronger.

**Author comment:**

We agree that the reference is a little outdated to represent recent estimates. Therefore, we have revised the sentence as you suggested (L672~L674).

**Referee#1's comment:**

Specific minor comments

L11. ", also" to "is based on X, and has been…" At the moment it sounds like ICEFLUX is also in ECMWF model but I don't believe that's what you mean. And you need to say what the underlying input variables of the ECMWF parametrisation are

**Author comment:** We have rewritten this sentence to make it clear (L10~L14).

**Referee#1's comment:**

L19. "observations" of what?

**Author comment:** We have clarified this point in our revised manuscript (L22).

**Referee#1's comment:**

L30. "the reproductivity of long-term trends of lightning" to "trends of lightning over longer time periods" or something similar

**Author comment:** We have revised the sentence as you suggested (L34~L35).

**Referee#1's comment:**

Introduction. I think somewhere (intro or discussion section maybe) it needs to be acknowledged that whilst ice-based parametrisations are appealing they have greater uncertainty associated with inputs, especially with regard to the microsphyics scheme used. This paper can provide a good basis for that https://agupubs.onlinelibrary.wiley.com/doi/full/10.1029/2021JD035461?casa_token=XPjZ7-BBFZEAAAAA%3AfbdQYuwknvDDXqNO5TJGvAEiUNFpzLMnbt2VnnKcHMC4QXtaq1hUzOs3-PYNAOZvSowW3J3vLuvB

**Author comment:**

Yes, we agree. We have added a description and relevant reference in L633~L636 as you recommended.

**Referee#1's comment:**

L50. I think it would be worth referencing this paper, regarding what chemistry models are currently doing. It is much more up to date (though no updates to lightning schemes) and it includes some nice looks at the chemistry/radiative impacts of lightning in models. https://acp.copernicus.org/articles/21/1105/2021/

**Author comment:**

We have added the relevant content and the reference in L55~L57 as you suggested.

**Referee#1's comment:**

L57. As in abstract you don't say what the input variables are for the lopez scheme.

**Author comment:**

We have added a description in L65~L67 to clarify this point.

**Referee#1's comment:**

L76. A decrease for convective mass flux scheme was also found for CMAM model in Finney (2016a). Not necessary to include but would demonstrate it wasn't only one model in which this has been seen.

**Author comment:** Thank you for your information. We agree with you.

**Referee#1's comment:**

L72. You refer to a USA specific result but actually the tropics is where the dominant $LNO_x$ production occurs. The following

paper considers a cloud ice based parametrisation for lightning of Africa and finds a relatively small response of lightning to climate change over the continent https://agupubs.onlinelibrary.wiley.com/doi/full/10.1029/2020GL088163

**Author comment:** The paper you mentioned is interesting and we have added the findings of this paper in L84~L86 in our revised manuscript.

**Referee#1's comment:**

L139. I don't see that Q_R is a "flux" as there's not per second in there. Please include the units. I think its kg^2/kgm^2? A bit unusual so not sure how to describe it. You might almost have a flux with Q_R*sqrt(CAPE) as the sqrt of CAPE has units of m/s. But still not sure it works out so I suggest coming up with more precise terminology for what Q_R is.

**Author comment:** Actually, the "frozen precipitation convective flux" at L139 in the preprint doesn't represent $Q_R$. The "frozen precipitation convective flux" is $P_f$ used in equations (9) and (10) in the preprint. We have added the unit of $Q_R$ at L171 in our revised manuscript and the $Q_R$ is defined in L171~L175 as proposed by Lopez (2016).

**Referee#1's comment:**

L140. Can you give a reason for the "1.8" in the min function? What are the units of that? Km?

**Author comment:** As explained by Lopez (2016), the number 1.8 used in equation (6) is a constraint to let the term $z_{base}$ remains constant after it exceeds 1.8 km. Please note that based on the suggestion from the Referee#3's comment, the equation (6) in our revised version is now standardized on base SI units.

**Referee#1's comment:**

L170. Eq13 is quite different to Eq8, and changes the meaning of this variable quite a bit. For the better potentially – I can at least see the logic behind it more clearly. I'm not sure I'd call this a modification of the ECMWF param, I'd say it's probably different enough to have its own identity.

**Author comment:** Thank you for your suggestion. We now re-named this lightning scheme the ECMWF-McCAUL scheme since it is based on the original ECMWF scheme and McCaul's study (McCaul et al., 2009).

**Referee#1's comment:**

Table 1. Please review your use of "frozen precipitation convective flux". I do not think it makes sense. Stick to the variables present in the main equation for the parametrisation. ECMWF-mod (Eq 11/12) is based on column precipitating ice and CAPE so I think you should only have two bullet points. I also don't think it's a appropriate term for the ECMWF-original inputs but as per L139 comment I'm not really sure what you'd call that.

**Author comment:** Actually, the "frozen precipitation convective flux" is $P_f$ used in equations (9) and (10) in the preprint. We have now revised Table 1 and L161 as you recommended.

**Referee#1's comment:**

L186. What is the C-shaped scheme, Pickering et al.? reference?

L238. "profile as per Ott(2010)". Add reference and suggest just get rid of L186-190 which is confusing as it sounded like you were suggesting you'd explored both profiles. If you have then it would be good to know what you found.

**Author comment:** We have gotten rid of L186-190 as you recommended. We also added the reference at L271.

**Referee#1's comment:**

Table 2. Are experiments 3-6 actually different to experiment 3 or have you just selected out the relevant years? If they are different you need to say how, if not then it seems overkill to call them separate experiments.

**Author comment:** Experiments 4-6 are different from experiment 3. For experiment 3, the 10-years (2011-2020) mean $LNO_x$ production was adjusted to 5.0 TgN $yr^{-1}$ but this does not mean that the $LNO_x$ production in each year (e.g., 2016) is exactly 5.0 TgN $yr^{-1}$. However, for experiments 4-6 the annual global $LNO_x$ production was exactly adjusted to 5.0 TgN $yr^{-1}$. We have more explicitly clarified this in L286~L291.

**Referee#1's comment:**

Section 2.3.2. I presume you have sampled the specific flight track and timings from the modelled data, for comparison to observations? Can you explicitly state that.

**Author comment:** Yes, we have sampled the specific flight track and timings from the modelled data. We have now explicitly stated that in L239~L240.

**Referee#1's comment:**

Fig3. Showing separate diagrams for full global, ocean-only and land-only could draw out your points about the differences. All schemes, currently lie quite close to each other.

**Author comment:** We thank your suggestion and we have now revised Fig3 to include ocean-only and land-only Taylor diagrams.

**Referee#1's comment:**

L365. Is this model bias the total global model bias? Or just over the regions in fig9? Is it on an annual or monthly basis?

**Author comment:** For L365 in the preprint, the model bias is the total annual global model bias. We have now specifically stated that at L454.

**Referee#1's comment:**

L370-379 and L548. The differences between these correlation coefficients are generally quite small, and the differences probably not significant. I would go easy on firm conclusions here and at least acknowledge that the correlations are actually very similar (this is something you should consider throughout when referring to differences in correlations).

**Author comment:** We agree. And we have now revised the sentence (L459~L460) as you suggested. For L548, we have conducted a significant test and found that the differences are significant even though the differences are small.

**Referee#1's comment:**

L437. The majority of these references are looking at the equilibrium response to warming. Clark had a transient simulation but I suspect averaged over a number of decades to show the trend. You are looking at variations over a much smaller timescale. It is interesting to see, but it is comparing apples with oranges to directly compare these. You should acknowledge that they are measuring different kinds of response, over different timescales. You would be better going back to the Williams paper that looks at temperature response on shorter timescales, and relating your results to that… https://ui.adsabs.harvard.edu/abs/2005AtmRe..76..272W/abstract

**Author comment:** Thanks for your suggestion. We have gotten rid of L437~L441 and added a discussion in L672~L674.

**Referee#1's comment:**

Fig12. Worth noting in caption why ICEFLUX doesn't have shading towards the poles?

**Author comment:** The ICEFLUX scheme doesn't have shading towards the poles because the upward cloud ice flux used is diagnosed as zero by the CHASER model typically within the high latitude regions. We have now specifically stated this in the caption of Fig13.

**Referee#1's comment:**

Technical corrections

L9. "improve" to "improving"

L65. "CPAE" to "CAPE"

Fig6, table3. Say clearly in the caption that these are looking at NO. Currently it doesn't state.

Fig 10. State in the caption that these are relative to the CTH simulation.

Fig11. Please say what the black blue and red lines are in the caption.

Fig14. Is this accessible to people with red green colour blindness?

**Author comment:** We are sorry for our careless mistakes, and it was rectified in L9, L76, Fig. 7 and table3, Fig. 11, Fig. 12, Fig. 15 and Fig. 16, respectively.

**Response to Anonymous Referee #2's comment,**

**Referee#2's comment:**

Four lightning $NO_x$ production schemes were investigated in the CHASER (MIROC) chemistry climate model: ICEFLUX scheme, the original and modified ECMWF schemes, and the CTH scheme (native to the CHASER model). The model performance was evaluated using OTD (lightning flashes), ATOM (air composition), TROPOMI ($NO_2$ and $O_3$ columns). As the most abundant $NO_x$ sources in the mid to upper troposphere, it is important to include $LNO_x$ in global models by implementing and testing $LNO_x$ production schemes. The manuscript is generally well written and clearly presented, but some important pieces were missing or not properly addressed. A major revision is needed before it can be published in GMD for the following reasons:

**Author comment:** Thanks for Referee#2's dedicated time. We also appreciate your interest in our manuscript and the overall positive assessment of our work.

**Referee#2's comment:**

1. In half of the simulations, the global $LNO_x$ emissions were adjusted to 5.0 TgN/yr using different adjustment factors, but no descriptions were given regarding what are the adjustment factors and how was it done? A detailed description is necessary to understand the process.

**Author comment:** In the case of the third set of experiments (2011-2020), temporally and spatially uniform adjustment factors were applied to adjust the mean $LNO_x$ production (2011–2020) to 5.0 TgN yr$^{-1}$. This adjustment was achieved by first conducting the simulations without any adjustment and the 2011–2020 mean $LNO_x$ production ($P_{LNO_x}$) was calculated, then the corresponding adjustment factor ($adj\_factor$) can be calculated by using the following equation.

$$adj\_factor = \frac{5.0}{P_{LNO_x}} \quad (1)$$

Similarly, we also adjusted the $LNO_x$ emissions in the fourth to the sixth sets of experiments to 5.0 TgN yr$^{-1}$.

We have now added a detailed description about the adjustment factors in L284~L291 in our revised manuscript. We hope that the added description can answer the questions about what the adjustment factors are and how was it done.

**Referee#2's comment:**

2. The NO emissions per flash were set to 111 moles NO per IC (intro-cloud lightning flash) and 1113 moles NO per CG (cloud-to-ground lightning flash) as parameters drawn from work reported by Price et al. (1997). These values were quite outdated, and there are many publications to update the lightning production efficiency (PE) in recent years. And the number 1113 is considered the upper limit of the reported values and is not realistic. In recent years, the commonly recommended values for CG flashes are in the range of 150-350. Please justify your use of these numbers. At the very least, please discuss the uncertainties that may be caused by using different PE values.

**Author comment:** We agree that the $LNO_x$ PE values (IC:111/CG:1113) are outdated and we have now updated these values in our revised manuscript to IC=CG=250 mol fl.$^{-1}$ based on the recent studies (Allen et al., 2019). As a result, we surprisingly found that the model biases (NO) against the ATom observations were significantly reduced by using the new $LNO_x$ PE values. We also discussed the uncertainties associated with different PE values in L273~L276. Thank you very much for your useful comments.

**Referee#2's comment:**

3. In the validation of the lightning schemes, the 1996-2000 OTD climatological data was used to compare with the simulations during 2007-2011. As a qualitative measure, it is probably okay to use the climatological data for a broad range comparison even though the time periods were completely separated. However, since the LIS data was available for the simulation years, albeit it was in a narrower latitude range, it should be used to evaluate the model's performance.

**Author comment:** We have now also used the LIS 2007-2011 lightning climatologies to evaluate the model's performance in Sect. 3.1.

**Referee#2's comment:**

4. In recent versions of the TROPOMI $NO_2$ column data (http://www.tropomi.eu/data-products/nitrogen-dioxide), there is increase of the tropospheric $NO_2$ columns under some weather conditions. Please provide the detailed version (and level) information of TROPOMI data used in this research and discuss potential uncertainties associated with it.

**Author comment:** Thanks for your suggestion. The TROPOMI data used in our research is level-2 offline (OFFL) tropospheric $NO_2$ columns in 2019. The product version is 1.0.0 from 2019-01-01 to 2019-03-20 and updated to 1.1.0 from 2019-03-21 to 2019-12-31. We have now provided the TROPOMI product information in L246~L247. From the official website of the TROPOMI $NO_2$ column data, it is explained that the upgrade to version 2.2.0 (5 July 2021) leads to an increase of the tropospheric $NO_2$ columns for cloud-free pixels. However, the TROPOMI data used in our study (1.0.0~1.1.0) do not involve that upgrade. It is reported that the TROPOMI data used in our study biased negatively compared to the airborne or ground-based observation data (Tack et al., 2021; Verhoelst et al., 2021; van Geffen et al., 2022). Therefore, we also discussed the potential uncertainties that could exist in the TROPOMI data in L644~L648.

**Referee#2's comment:**

5. The Taylor diagram showing the prediction accuracy of the various lightning schemes doesn't seem to present any significant differences among the schemes. With mean values across the globe and spanning so many years, correlation coefficients of 0.80 and 0.79 probably don't mean much difference. Please provide the significant test and, if possible, a confidential interval would help.

**Author comment:** We have now conducted a significant test to determine whether the differences in the statistics are significant. This part is added to our revised manuscript (L339~L355).

**Referee#2's comment:**

6. I would suggest replacing Figures 4 and 5 using LIS data with the same period as your simulation years. The peaks (most lightning occurrences) appeared between +38° and -38° latitude that the LIS had coverage anyway.

**Author comment:** Thanks for your suggestion and we now updated Fig. 4 and Fig. 5 as you suggested. Fig.4 and Fig.5 in the preprint changed to Fig.5 and Fig. 6 in our revised manuscript.

**Referee#2's comment:**

7. The trend analyses for lightning density (Section 3.4) and $NO_x$ and $O_3$ columns (Section 3.5) were only based on simulation results and compared to the CTH lightning scheme. Because no observational basis, all the analysis is hypothetical and there is no way to know which scheme or schemes could reproduce (or even close to) the truth. Understandably limitation exists in time and space to employ observation-based analysis, but it is still helpful if limited analysis based on observations could be done. For example, for the lightning trend, the LIS/OTD satellite data could be used from the years when the data was available to derive the climatological trend coupled with the trend of temperature during the period. While for $NO_2$ and $O_3$ columns, the OMI/TROPOMI data could be employed to examine the trend over the years.

**Author comment:** We agree. We have now provided the comparisons between the modelled tropospheric columns ($NO_2$ and $O_3$) and the OMI tropospheric column observations (please refer to Fig. S1 and Fig. S2 in the newly added supplement of the manuscript). For the lightning trends, we also attempted to use the LIS/OTD satellite observations to evaluate the simulated lightning trends. However, the lightning trend (2001-2013) derived from the LIS/OTD observations is almost flat with no significant trend determined by the Mann–Kendall rank statistic. Therefore, we considered it is inappropriate to use the lightning trend (2001-2013) based on LIS/OTD observations to examine the simulated lightning trends. We expect that the lightning observations for a longer period can be used to verify the simulated lightning trends in future research. But if the Editor or Referees think it is better to add this part, we can add it to our manuscript.

**Referee#2's comment:**

Specific Comments:

1. Page 1 Line 8 and other places, "the most dominant $NO_x$ source in the upper troposphere", I would say mid to upper troposphere.

**Author comment:** We have now revised the sentence as you suggested (L8 and L42).

**Referee#2's comment:**

2. Page 2, Lines 40-47, not only in global models, the studies of $LNO_x$ in regional scale models have also made significant progress in recent years (e.g. Kang et al. 2019: https://doi.org/10.5194/gmd-12-3071-2019 and https://doi.org/10.5194/gmd-12-4409-2019 and 2020: https://www.nature.com/articles/s41612-020-0108-2), please update the knowledge with references in the discussion.

**Author comment:** Thanks for providing the up-to-date information about the studies of $LNO_x$ in regional-scale models. We are glad to see that the $LNO_x$ studies in regional-scale models have also made great progress in recent years. We have now updated the knowledge in L57~L58 as you suggested.

**Referee#2's comment:**

3. Page 2, Lines 51-55, "However, some new … against the LIS/OTD lightning observations". These two sentences are basically stating the same thing, please combine and simplify these sentences.

**Author comment:** We have combined and simplified these sentences as you suggested (L61~L64).

**Referee#2's comment:**

4. Page 2, Lines 57-58, "The two new lightning schemes …", please provide references.

**Author comment:** We have now provided the references (L68)

**Referee#2's comment:**

5. Page 2, Line 58, "The new lightning schemes must be validated …", the expression "must be" is too strong, consider rewording.

**Author comment:** We agree, and we have now revised it to "are expected to be evaluated and compared…" at L69.

**Referee#2's comment:**

6. Page 19, end of Figure 14 caption, the "without nudging" used twice. One of them should be "with nudging".

**Author comment:** We are sorry for the careless mistake, and we have rectified it at L597.

**Referee#2's comment:**

7. Page 21, Line 543 and other locations, "Two new lightning schemes". I would say Three new lightning schemes: the ICEFLUX, two ECMWF schemes (original and modified). It would be more straightforward to me to just say 3 new schemes.

**Author comment:** We appreciate this good suggestion, and we have revised the sentence (L629) as you suggested.

**Referee#2's comment:**

8. One comment on the ICEFLUX as it displayed the downward trend of lightning flashes with rising temperature. Since the rising temperature would mean less ICE and less ICE FLUX, that would result in fewer lightning flashes. This probably suggests the intrinsic shortcomings for this scheme to simulate climate lightning trends.

**Author comment:** We agree with you that the rising temperature would mean less ICE and less ICE FLUX, which would result in fewer lightning flashes. But we think that further researches are needed to clarify whether this suggests the intrinsic shortcomings of this scheme or not. Please note that we have also introduced the isotherm-alternative approach for the ICEFLUX scheme to the CHASER (MIROC) chemistry-climate model as suggested by Referee#1's comment. We have added the results from the isotherm-alternative approach in our revised manuscript (L153~L158; Sect. 3.4 and Sect. 3.5; newly added supplement Fig. S1 and Fig. S2)

**Reply on RC2 - regarding specific point 8', Declan Finney:**

I am largely in agreement with the reviewer's comments, except with regard to their specific point 8:

"One comment on the ICEFLUX as it displayed the downward trend of lightning flashes with rising temperature. Since the rising temperature would mean less ICE and less ICE FLUX, that would result in fewer lightning flashes. This probably

suggests the intrinsic shortcomings for this scheme to simulate climate lightning trends."

I think whether there will be less cloud ice (in the column) in a future climate could be a more complex question than is implied in this statement, but that is not my main concern. I can go along with expecting less cloud ice at a given tropospheric pressure level due to warming (though there are possibly caveats even to that regarding e.g. aerosols).

My main issue, is implying that the use of cloud ice itself is an intrinsic shortcoming. If it's accepted that the non-inductive charging mechanism is an appropriate basis for a lightning parametrisation, then the implication is that in future if cloud ice (and cloud ice fluxes) reduce then electrical charging will reduce too. This provides a line of scientific reasoning to explain why lightning might reduce in future (whether we can reliably simulate the change in cloud ice in future is another matter).

The shortcoming/challenge of the ICEFLUX parametrisation is not its use of cloud ice, but its application of a pressure level. If seen as a challenge then one could take the approach of Finney et al. (2018) and adjust the pressure level in response to the changing climate. Or if seen as a shortcoming then I suggest the isotherm variation proposed by Romps (2019, http://dx.doi.org/10.1029/2019GL085748) is well worth exploring.

I recommend including a paragraph highlighting the above modelling considerations with regard to the ICEFLUX parametrisation and climate change simulations.

**Author comment:** Thank you very much for sharing your insights about why lightning may reduce in the future. We have now included a paragraph to highlight the above modelling considerations in L667~L670 as you recommended.

**Response to Anonymous Referee #3's comment,**

**Referee#3's comment:**

General Comments

This paper presents an interesting set of model experiments that tests different representations of lightning. Unfortunately it is rather unclearly written and many important details are missing, so I have struggled to fully assess its contents. Some of this lack of clarity and missing details are listed below in the Specific Comments section. For example, it is unclear what the un-nudged simulations represent, and it is hence difficult to interpret what is meant by the difference between the nudged and un-nudged runs (and these differences are large). Other aspects of the experiments are also not very clear – e.g., how is soil $NO_x$, the other major natural $NO_x$ source, handled? Presumably it is somehow fixed. It is also unclear how the other major $NO_x$ emissions (anthropogenic and biomass burning) are prescribed in the experiments. Some of the units are incorrect or unclear. There is little detail about how the TROPOMI comparison is carried out, and I am unsure from what is written if averaging kernels have been applied correctly, to make this a valid evaluation. I appreciate this is mainly a paper that describes the lightning schemes and their differences, but it would be nice to include a bit more discussion on the interpretation of the differences, particularly as they appear to lead to very different trends in lightning. If all these points can be clarified in a revised version, then this paper may be acceptable for GMD, but in its current form it is not.

**Author comment:**

We sincerely thank the reviewer for your time and devotion to pointing out our manuscript's shortcomings. To improve the manuscript, we will specifically clarify each point. We explicitly pointed out where the manuscript is revised in this response letter.

**Referee#3's comment:**

Specific Comments

L8 most dominant -> dominant

**Author comment:** We have now modified the sentence at L8 and L41 as you suggested.

**Referee#3's comment:**

L11-12 Sentence does not make sense

**Author comment:** We have revised the sentence (L11~L14).

**Referee#3's comment:**

L19 Biases for what?

**Author comment:** We have specifically clarified this point in our revised manuscript (L22).

**Referee#3's comment:**

L27 Why does nudging make such a large impact on the trends? What do the un-nudged runs represent?

**Author comment:** Since nudging toward meteorological reanalysis data cannot be used when predicting lightning trends under future climate changes, we also chose to show the results without nudging. This can update our knowledge about the sensitivity of simulated lightning trends to meteorological nudging. The un-nudged runs also represented the short-term surface warming like the experiments with nudging. The only differences between the un-nudged and nudged experiments are whether the meteorological fields (u,v,T) in the model are nudged towards the six-hourly NCEP FNL data. Meteorological nudging will lead to significant changes of the meteorological fields and the diagnosed proxies (e.g., CAPE) used to compute lightning densities. Therefore, the meteorological nudging can make a large impact on simulated lightning trends. We have now specifically clarified this point in our revised manuscript (L30, L523~L527, L537~L544).

**Referee#3's comment:**

L27 Was the earlier study also 2001-2020 changes?

**Author comment:** The earlier study mentioned here was not 2001-2020 changes. Therefore, we have gotten rid of this part (L29).

**Referee#3's comment:**

L30 reproducibility?

**Author comment:** We have now changed to "reproducibility" (L34).

**Referee#3's comment:**

L35 Is there such a thing as unreactive NO?

**Author comment:** Thanks for your comment. We have gotten rid of the word "reactive" (L39).

**Referee#3's comment:**

L37 Finney et al (also l53)

**Author comment:** We are sorry for our careless mistakes. We have now revised them (L40, L62).

L57 The schemes are not 'described above'

**Author comment:** We have now added the reference of the two lightning schemes and changed the "described above" to "mentioned above" (L68).

L57 I much prefer evaluated to "validated"

**Author comment:** Thank you for your good suggestion, and we have revised the word (L68).

L66 CAPE – also clarify "CAPE x precipitation as a proxy"

**Author comment:** We have revised the word "CAPE" and clarified "CAPE x precipitation as a proxy" (L76).

L72 Contiguous?

**Author comment:** We have revised this point (L82).

L86 "This study" is ambiguous

**Author comment:** We have changed "This study" to "Our study" (L98).

L91 (etc) Capitalise Section references

**Author comment:** We have capitalized the Section references (L105~L109).

L118 What about other natural $NO_x$ sources, e.g. soil?

**Author comment:** The monthly soil $NO_x$ emissions used in CHASER (MIROC) are constant for each year and are derived from Yienger and Levy (1995). We have specifically stated this point (L131~L132).

L132 over land and ocean

**Author comment:** We have revised "of land and ocean" to "over land and ocean" (L146).

L133 Check units

**Author comment:** We are sorry for the mistake, and we have revised the units (L147).

L135 Why was 440 hPa chosen?

**Author comment:** The 440 hPa pressure level is chosen because it is a representative pressure level of fluxes in deep

convective clouds (Finney et al, 2014). We have now specifically explained it in L150~L151.

**Author comment:** The levels between the 0° and -25°C isotherms were specified for the frozen precipitation flux, which was explained in equation 8 ~ equation 10. Of course, we think the frozen precipitation flux varies hugely with vertical levels.

**Author comment:** We have now standardized all equations on base SI units as you suggested (L162~L192).

**Author comment:** Alpha is not dimensionless. Alpha has the units $(\text{fl.}\,\text{kg}^{-1}\,\text{m}^{-3})$. We have provided the units of alpha at L163.

**Author comment:** We agree, and we have provided the units of these alphas at L191.

**Author comment:** Since $z_0$ and $z_{-25}$ are already defined here, we have gotten rid of the definition of $z_0$ and $z_{-25}$ here (L198).

**Author comment:** We have now revised the sentence to clarify it (L215).

**Author comment:** We have capitalized Earth (L222).

**Author comment:** Since most of the lightning occurs over land regions during summer, ATom1 (July–August 2016) and ATom2 (January–February 2017) were used to evaluate $LNO_x$ emissions (corresponding to summer in the northern and southern hemispheres, respectively). We have specifically clarified in L234~L236.

L236 Table 2 could give more details of each experiment. E.g., are HTAP2 (2008) anthropogenic emissions used for all run years? Are the MACC GFAS biomass burning emissions annually varying? What is done with soil $NO_x$ emissions? Is nudging off equivalent to a free running climate?

**Author comment:** We have revised Table 2 to give more information about each set of experiments.

**Referee#3's comment:**

L245 Why compare OTD data from 1996-2000 with simulations of 2007-2011? Are these simulations with nudging on or off?

**Author comment:** We compare OTD data (1996-2000) with simulations (2007-2011) because the comparison can be conducted in a wider range (±75° latitude). We have also evaluated the potential uncertainties associated with the inconsistency of the time period of simulated lightning and observed lightning (L305~L309). These simulations were conducted with nudging, and we have explicitly pointed it out (L304).

**Referee#3's comment:**

L353 Did you sample the model at TROPOMI overpass times, and then somehow produce the monthly mean values in Figure 9? Please clarify how you applied the averaging kernels to demonstrate that the model-satellite comparison is like-for-like.

**Author comment:** Yes, we have sampled the model at TROPOMI overpass times. We have clarified how we applied the averaging kernels in L247~L257 and L437~L443.

**Referee#3's comment:**

L394 Clarify the plots in Figure 10 are all relative to CTH.

**Author comment:** We have clarified at L504.

**Referee#3's comment:**

L422 Figure 11 I am unclear what the un-nudged runs represent. Do these runs somehow represent climate change over the 2001-2020 period, or are they completely unforced?

**Author comment:** As we have explained above. Since nudging to meteorological reanalysis data cannot be used when predicting lightning trends under future climate changes, we also chose to show the results without nudging. This can update our knowledge about the sensitivity of simulated lightning trends to meteorological nudging. The un-nudged runs also represented the short-term surface warming due to the forcing with GHGs concentrations and SSTs and Sea ice (please refer to Table 2 in our revised manuscript) like the experiments with nudging. The only differences between the un-nudged and nudged experiments are whether the meteorological fields are nudged towards the six-hourly NCEP FNL data.

**Reference used by author's comment:**

Allen, D. J., Pickering, K. E., Bucsela, E., Krotkov, N., and Holzworth, R.: Lightning $NO_x$ Production in the Tropics as Determined Using OMI $NO_2$ Retrievals and WWLLN Stroke Data, J. Geophys. Res. Atmos., 124, 13498–13518, https://doi.org/10.1029/2018JD029824, 2019.

Finney, D. L., Doherty, R. M., Wild, O., Huntrieser, H., Pumphrey, H. C., and Blyth, A. M.: Using cloud ice flux to parametrise large-scale lightning, 14, 12665–12682, https://doi.org/10.5194/acp-14-12665-2014, 2014.

Lopez, P.: A lightning parameterization for the ECMWF integrated forecasting system, 144, 3057–3075, https://doi.org/10.1175/MWR-D-16-0026.1, 2016.

McCaul, E. W., Goodman, S. J., LaCasse, K. M., and Cecil, D. J.: Forecasting lightning threat using cloud-resolving model simulations, Weather Forecast., 24, 709–729, https://doi.org/10.1175/2008WAF2222152.1, 2009.

Tack, F., Merlaud, A., Iordache, M. D., Pinardi, G., Dimitropoulou, E., Eskes, H., Bomans, B., Veefkind, P., and Van Roozendael, M.: Assessment of the TROPOMI tropospheric $NO_2$ product based on airborne APEX observations, Atmos. Meas. Tech., 14, 615–646, https://doi.org/10.5194/amt-14-615-2021, 2021.

van Geffen, J., Eskes, H., Compernolle, S., Pinardi, G., Verhoelst, T., Lambert, J.-C., Sneep, M., ter Linden, M., Ludewig, A., Boersma, K. F., and Veefkind, J. P.: Sentinel-5P TROPOMI $NO_2$ retrieval: impact of version v2.2 improvements and comparisons with OMI and ground-based data, Atmos. Meas. Tech., 15, 2037–2060, https://doi.org/10.5194/amt-15-2037-2022, 2022.

Verhoelst, T., Compernolle, S., Pinardi, G., Lambert, J. C., Eskes, H. J., Eichmann, K. U., Fjæraa, A. M., Granville, J., Niemeijer, S., Cede, A., Tiefengraber, M., Hendrick, F., Pazmiño, A., Bais, A., Bazureau, A., Folkert Boersma, K., Bognar, K., Dehn, A., Donner, S., Elokhov, A., Gebetsberger, M., Goutail, F., Grutter De La Mora, M., Gruzdev, A., Gratsea, M., Hansen, G. H., Irie, H., Jepsen, N., Kanaya, Y., Karagkiozidis, D., Kivi, R., Kreher, K., Levelt, P. F., Liu, C., Müller, M., Navarro Comas, M., Piters, A. J. M., Pommereau, J. P., Portafaix, T., Prados-Roman, C., Puentedura, O., Querel, R., Remmers, J., Richter, A., Rimmer, J., Cárdenas, C. R., De Miguel, L. S., Sinyakov, V. P., Stremme, W., Strong, K., Van Roozendael, M., Pepijn Veefkind, J., Wagner, T., Wittrock, F., Yela González, M., and Zehner, C.: Ground-based validation of the Copernicus Sentinel-5P TROPOMI $NO_2$ measurements with the NDACC ZSL-DOAS, MAX-DOAS and Pandonia global networks, Atmos. Meas. Tech., 14, 481–510, https://doi.org/10.5194/amt-14-481-2021, 2021.

Sincerely,
On behalf of all co-authors,
Yanfeng He.

---

## Author Response (AR2)

**Response to Referee#1's comment,**

**Referee#1's comment:**

The authors have done a good job of revising the paper. I take the following slight issue which I suggest should have a minor rewording. Otherwise, I consider the manuscript ready to publish.

The authors have included good additional analysis of ICEFLUX_T, which I think is very helpful.

However, I feel the following sentence does not quite frame ICEFLUX_P correctly.

L533 - "As explained by Romps (2019), the ICEFLUX_P approach is based on a fixed isobar which makes it inappropriate for climate change studies."

ICEFLUX_P is not "inappropriate" for climate change studies. Finney et al (nature climate change, 2018) demonstrated how it could be applied to a future climate. However, the use of an isotherm is probably more transparent and less involved. There's different ways the text could be modified, but a suggestion would be to replace "inappropriate" with "less convenient". Or you could say "the parametrisation would require modification to allow the isobar level to adapt to the changing climate".

**Author comment:**

We are thankful for your patient inspection of our revised manuscript. We are pleased to see the revised manuscript has met your satisfaction. We agree that the description of ICEFLUX_P scheme here is not accurate and we have revised the word "inappropriate" to "less convenient" at L534 as you suggested.

Sincerely,

On behalf of all co-authors,

Yanfeng He.